


# Understanding the Mass, Momentum and Energy Transfer in the Frozen Soil with Three Levels of Model Complexities

Lianyu Yu[1], Yijian Zeng[1], Zhongbo Su[1,2]

[1] Faculty of Geo-information and Earth Observation (ITC), University of Twente, Enschede, The Netherlands

[2] Key Laboratory of Subsurface Hydrology and Ecological Effect in Arid Region of Ministry of Education, School of Water and Environment, Chang'an University, Xi'an, China

*Correspondence to*: Yijian Zeng (y.zeng@utwente.nl); Zhongbo Su (z.su@utwente.nl)

## Abstract

Frozen ground covers vast area of earth surface and has its important ecohydrological implications for high latitude and high altitude regions under changing climate. However, it is challenging to characterize the simultaneous transfer of mass and energy in frozen soils. Within the modeling framework of STEMMUS (Simultaneous Transfer of Mass, Momentum and Energy in Unsaturated Soil), the model complexity of soil heat and mass transfer varies from uncoupled, to coupled heat and mass transfer, and further to the explicit consideration of airflow (termed as unCPLD, CPLD, and CPLD-AIR, respectively). The impact of different model complexities on understanding the mass, momentum and energy transfer in frozen soil were investigated. The model performance in simulating water and heat transfer and surface latent heat flux was tested on a typical Tibetan Plateau meadow. Results indicate that the CPLD model considerably improved the simulation of soil moisture, temperature and latent heat flux. The analyses of heat budget reveal that the improvement of soil temperature simulations by CPLD model is ascribed to its physical consideration of vapor flow and thermal effect on water flow, with the former mainly functions above the evaporative front and the latter dominates below the evaporative front. The contribution of airflow-induced water and heat transport to the total mass and energy fluxes is negligible. Nevertheless, given the explicit consideration of airflow, vapor flow transfer and its effect on heat transfer were enhanced during the freezing-thawing transition period.





## 1. Introduction

Frozen soils, have been reported with significant changes under climate change (Cheng and Wu, 2007;Hinzman et al., 2013;Zhao et al., 2019). Changes in freezing/thawing process can alter soil hydrothermal regimes, activate/close the water flow pathways and vegetation development (Walvoord and
Kurylyk, 2016). Such changes will further considerably affect the spatial pattern, the seasonal to interannual variability and long term trends in land surface water, energy and carbon budgets and then the land surface atmosphere interactions (Schuur et al., 2015;Subin et al., 2013;Walvoord and Kurylyk, 2016). Understanding the soil freeze/thaw processes appears to be the necessary path to the better water resources management and ecosystem protection in cold regions.

When soil experiences the freeze/thaw process, there is a dynamic thermal equilibrium system of ice, liquid water, water vapor and dry air in soil pores. Water and heat flow are tightly coupled in frozen soils. Coupled water and heat physics, describing the concurrent flow of liquid, vapor as well as heat flow, was first proposed by Philip and De Vries (1957), (hereafter termed as PdV57) considering the enhanced vapor transport. The PdV57 theory has been widely applied for detailed understanding of soil evaporation during the drying
process (De Vries, 1958;De Vries, 1987;Milly, 1982;Novak, 2010;Saito et al., 2006). The attempts to simulate the coupled water and heat transport in frozen soils started in 1970s (e.g., Guymon and Luthin, 1974;Harlan, 1973). Since then, numerical tools able to and subjected to simulate one dimensional frozen soil were increasingly developed. Flerchinger and Saxton (1989) developed the SHAW model with the capacity of simulating the coupled water and heat transport process. Hansson et al. (2004) accounted for the
phase changes in HYDRUS-1D model and verified its numerical stability with rapidly changing boundary conditions. Considering the two components (water and gas) and three water phases (liquid, vapor, and solid), Painter (2011) developed a fully coupled water and heat transport model MarsFlo. These works together with other modifications, simplifications, generate a series of hierarchy of frozen soil models (detail reviewed by Li et al., (2010) and Kurylyk and Watanabe (2013)).

Air flow has been reported important to the soil water and heat transfer process under certain conditions (Prunty and Bell, 2007;Touma and Vauclin, 1986). Zeng et al., (2011a, b) found that soil evaporation is enhanced after precipitation events by considering air flow and demonstrated that the air pressure induced advective fluxes inject the moisture into the surface soil layers and increase the hydraulic conductivity at top layer. The diurnal variations of air pressure resulted in the vapor circulation between the atmosphere and land
surface. Wicky and Hauck (2017) reported that the temperature difference between the upper and the lower part of a permafrost talus slope was significant and attributed it to the airflow induced convective heat flux. Yu et al., (2018) analyzed the spatial and temporal dynamics of air pressure induced fluxes and found an interactive effect as the presence of soil ice. The abovementioned studies demonstrate that the explicit consideration of air flow has the potential to affect the soil hydrothermal regime.





Current land surface models (hereafter LSMs), however, usually adopted a simplified frozen soil physics with relative coarse vertical discretization (Koren et al., 1999;Niu et al., 2011;Swenson et al., 2012;Viterbo et al., 1999). In their parameterizations, soil water and heat interactions can only be indirectly activated by the phase change processes, the mutual dependence of liquid water, water vapor, ice and dry air in soil pores is of course absent. This mostly lead to unrealistic physical interpretations and worse performance regarding

to the hydrothermal, ecohydrological dynamics (Cuntz and Haverd, 2018;Novak, 2010;Wang and Yang, 2018). Specifically, Su et al. (2013) evaluated the European Centre for Medium-Range Weather Forecasts (ECMWF) soil moisture analyses on the Tibetan Plateau, with HTESSEL as the land surface modelling component. Their results indicated the deficiency of HTESSEL in capturing phase transitions. How and to what extent the complex mutual dependence physics affects the soil mass and energy transfer in frozen soils?

Is it necessary to consider such a fully physical mechanism in LSMs? These two questions frame the scope of this work.

In this paper, we incorporated the various complexity of water and heat transport mechanisms into a common modeling framework (STEMMUS-FT, Simultaneous Transfer of Energy, Momentum and Mass in Unsaturated Soils with Freeze-Thaw). With the aid of in situ measurements collected from a typical Tibetan

meadow site, the pros and cons of different model complexities were investigated. Subsurface energy budgets and latent heat flux density analyses were further conducted to illustrate the underlying mechanisms considering different coupled water-heat physics. Section 2 describes the experimental site and the implementation of increasing complexity of subsurface physics into STEMMUS framework. Performance of different models is presented in Section 3 together with the subsurface heat budgets and latent heat flux

density analyses. Section 4 discusses the effects of considering coupled water-heat transport and air flow in frozen soils. Conclusion is made in Section 5.

## 2. Methodology

### 2.1 Experimental site

Maqu station, equipped with a catchment scale soil moisture and soil temperature (SMST) monitoring

network and micro-meteorological observing system (Dente et al., 2012;Su et al., 2011;Zeng et al., 2016), is situated on the north-eastern fringe of the Tibetan Plateau (33°30'–34°15'N, 101°38'–102°45'E). According to the updated Köppen-Geiger climate classification system, it can be characterized as a cold climate with dry winter and warm summer (Dwb). The mean annual air temperature is 1.2 ℃, and the mean air temperatures of the coldest month (January) and warmest month (July) are about -10.0 ℃ and 11.7 ℃,

respectively. Alpine meadows (e.g., *Cyperaceae* and *Gramineae*), with heights varying from 5 cm to 15 cm throughout the growing season, are the dominant land cover in this region. The sandy loam and silt loam are found by in situ soil sampling and organic soil with a maximum of 18.3 % organic matter for the upper soil layers (Dente et al., 2012;Zhao et al., 2018;Zheng et al., 2015).


The Maqu SMST monitoring network spans an area of approximately 40 km×80 km with the elevation ranging from 3200 m to 4200 m a.s.l. SMST profiles are automatically measured by 5TM ECH$_2$O probes (METER Group, Inc., USA) installed at the soil depths of 5 cm, 10 cm, 20 cm, 40 cm, and 80 cm. The micro-meteorological observing system includes a 20 m Planetary Boundary Layer (PBL) tower providing the meteorological measurements at five heights above ground (i.e., wind speed and direction, air temperature and relative humidity) , and an eddy-covariance (EC150, Campbell Scientific, Inc., USA) system installed for measuring the turbulent sensible, latent heat fluxes and carbon fluxes. Four component down and upwelling solar and thermal radiation (NR01-L, Campbell Scientific, Inc., USA), and liquid precipitation (T200B, Geonor, Inc., USA) are also monitored.

**2.2 Mass and energy transport in unsaturated soils**

On the basis of STEMMUS modelling framework, the increasing complexity of vadose zone physics in frozen soils was implemented as three alternative models (Table 1). Firstly, STEMMUS enabled the isothermal water and heat transfer physics (Eqs. 1 & 2). The 1-D Richards equation is utilized to solve the isothermal water transport in variably saturated soils. The heat conservation equation took into account the freezing/thawing process and the latent heat due to water phase change. The effect of soil ice on soil hydraulic and thermal properties was considered. It is termed as unCPLD model.

Secondly, the fully coupled water and heat physics, i.e., water vapor flow and thermal effect on water flow, was explicitly considered in STEMMUS, termed as CPLD model. For the CPLD physics, the extended version of Richards (1931) equation with modifications made by Milly (1982) was used as the water conservation equation (Eq. 3). Water flow can be expressed as liquid and vapor fluxes driven by temperature gradient and matric potential gradient, respectively. The heat transport in frozen soils mainly includes: heat conduction (CHF, $\lambda_{eff}\frac{\partial T}{\partial z}$ ), convective heat transferred by liquid flux (CFL, $-C_L q_L(T - T_r)$, $-C_L S(T - T_r)$), vapor flux (CFV, $-[L_0 q_V + C_V q_V(T - T_r)]$), the latent heat of vaporization (LHF, $\rho_V \theta_V L_0$), the latent heat of freezing/thawing ($-\rho_i \theta_i L_f$) and a source term associated with the exothermic process of wetting of a porous medium (integral heat of wetting) ($-\rho_L W \frac{\partial \theta_L}{\partial t}$). It can be expressed as Eq. 4 (De Vries 1958; Hansson et al. 2004).

Lastly, STEMMUS expressed the freezing soil porous media as the mutual dependence system of liquid water, water vapor, ice water, dry air and soil grains, in which the air flow was independently considered while the other keep the same as CPLD model, termed as CPLD-AIR model (Eqs. 5, 6, &7, Zeng et al. 2011a,b; Zeng and Su, 2013). The air flow induced water and vapor fluxes ($q_{La}$, $q_{Va}$) and its corresponding convective heat flow (CFa, $q_a C_a(T - T_r)$) were involved in the water and heat transfer mechanisms, respectively.



To accommodate the specific conditions of a Tibetan meadow, the total depth of soil column was set as 1.6 m. The vertical soil discretization was designed finer in the upper soil layers (0.1-2.5cm for 0-40cm) than that in the lower soil layers (5-20cm for 40-160cm). Three aforementioned models adopted the same set of

soil parameters, shown as Table 2.

## 3. Results

Given by the same atmospheric forcing and same set of parameters, the performance of models with various complexity of unsaturated soil water and heat physics was illustrated as Sect. 3.1, 3.2 & 3.3. Sect. 3.4 & 3.5 further analyzed the variations of heat budgets and subsurface latent heat flux density, intended to present

the underlying differences among various models.

### 3.1 Soil hydrothermal profile simulations

The performance of model with various soil physics in simulating the soil thermal profile information is illustrated in Fig. 1. Both CPLD and CPLD-AIR model well reproduced the time series of soil temperature at different soil depth except for the 40cm, which is probably due to the inappropriate measurements (e.g.,

improper placement of sensors). However, there are significant discrepancies of soil temperature simulated by the unCPLD model. Compared to the observations, a stronger diurnal behavior of soil temperature in response to the fluctuating atmospheric forcing was found and the earlier stepping in/stepping out of the frozen period was reproduced by the unCPLD model. Such differences enlarged at deeper soil layers with large BIAS and RMSE values (Table 3).

Figure 2 presents the time series of observed and model simulated soil liquid water content at five soil layers. During the rapid freezing period, a noticeable overestimation of diurnal fluctuations and early and fast decreasing of soil liquid water content was simulated by unCPLD model. Moreover, the strong diurnal fluctuations and early increase of liquid water content were also found during the thawing period. The early thawing of soil water even lead to an unrealistic refreezing process at 80 cm (from 88[th] to 92[th] day after

December 2015), which is due to the simulated early warming of soil by unCPLD model (Fig. 1). Such discrepancies were significantly ameliorated from CPLD and CPLD-AIR simulations. Nevertheless, all three models can well capture the diurnal variations and magnitude of liquid water content during the frozen period. Note that there is an observable difference between CPLD and CPLD-AIR simulated soil liquid water content at shallower soil layers during the thawing process (e.g., Fig. 2, 5cm).

### 3.2 Freezing front propagation

The time series of freezing front propagation derived from the measured and model simulated soil temperature was reproduced as Figure 3. Initialized from the soil surface, the freezing front quickly develops downwards till the maximum freezing depth. The thawing process starts from both the top and bottom, mainly driven by the atmospheric heat and geothermal heat source, respectively. Such characteristics were well





captured by both the CPLD and CPLD-AIR model in terms of freezing rate, maximum freezing depth and
surface thawing process. While the unCPLD model tended to present a more fluctuated and rapid freezing
front propagation and a deeper maximum freezing depth which is early reached. The effect of atmospheric
heat source on soil temperature was overestimated by the unCPLD model as shown by the stronger diurnal
early onset of the thawing process.

**3.3 Surface Evapotranspiration**

The performance of model with different soil physics in reproducing the latent heat flux dynamics is shown
in Fig. 4. Compared to the observed LE, there is a significant overestimation of half-hourly latent heat flux,
which significantly degraded the overall performance using unCPLD model. The occurrence of such
overestimation was notably reduced using CPLD and CPLD-AIR model. While the general underestimation

of latent heat flux by the CPLD and CPLD-AIR model was found mostly during the freezing-thawing
transition period (Fig. 5b), when the soil hydrothermal states are not well captured (Fig. 1 &2).

The overestimation of surface evapotranspiration by unCPLD model was significant during the initial
freezing and freezing-thawing transition period (Fig. 5a, December & February). During the rapid freezing
period (January), unCPLD model presented a good match in the diurnal variation compared to the

observations. The monthly average diurnal variations were found to be well captured by CPLD and CPLD-
AIR models. Figure 5b shows the comparison of observed and model simulated cumulative surface
evapotranspiration. The overall overestimation of surface evapotranspiration by unCPLD model can be
clearly seen in Fig. 5b. Days at the initial freezing periods, with high liquid water content simulations,
accounted for more than 90% of the overestimation. The initial stage overestimation of surface

evapotranspiration was significantly reduced by CPLD and CPLD-AIR simulations. Slight underestimation
of cumulative surface evapotranspiration was simulated by CPLD and CPLD-AIR model with values of 3.98%
and 4.78%, respectively.

**3.4 Heat budgets**

Figure 6 shows the time series of the model simulated energy budget components at 5cm using unCPLD,

CPLD and CPLD-AIR during the freezing period (5th - 11th day after 1 December) and freezing-thawing
transition period (83th - 89th day after 1 December). For the unCPLD model, only the rate of change of heat
content HC and conductive heat flux divergence CHF are considered as the LHS and RHS of Eq. 2. Three
additional terms, convective heat flux divergence of liquid flow HFL and vapor flow HFV, and latent heat
flux divergence were included for the CPLD model. While for the CPLD-AIR model, the convective heat

flux divergence of air flow HFa was further added. There is a strong diurnal variation of heat budget
components (HC, CHF & LHF), corresponding to the diurnal fluctuation of soil temperature. For the unCPLD
model, the rate of change of heat content is almost completely balanced by the conductive heat flux
divergence CHF, indicating an acceptably accurate simulation. Compared to the unCPLD model, a stronger
diurnal fluctuation of HC and CHF, characterized as larger maximum/minimum heat budget component



values, was found in CPLD model results. Rendered from results of Fig. 1, the time series of the first order
of soil temperature regarding to time ($\partial T/\partial t$) simulated by unCPLD model was larger than that simulated
by CPLD model. This indicates unCPLD model produced a series of less fluctuations of apparent heat
capacity term ($C_{app} = C_{soil} + \rho_i \frac{L_f^2}{gT} \frac{d\theta}{d\psi}$ ) than CPLD models. During the freezing period, the latent heat flux
divergence LHF was lower than conductive heat flux divergence CHF by 1-2 orders of magnitude. The

positive value of LHF term during daytime indicates condensation happens at 5cm, as water vapor moves
downward (see Yu et al. (2018)). The convective heat fluxes of liquid flow and vapor flow was even smaller
compared to conductive heat flux. There is no significant difference of heat budget components between
CPLD and CPLD-AIR model in terms of diurnal variation and magnitude. The convective heat flux
divergence of air flow played a negligible role on the change of thermal state (HC).

The dynamics of heat balance components at 5 cm soil layer was tested for the freezing-thawing transition
period (Fig. 6 d, e, f). Both HC and CHF underwent strong diurnal variations with increasing fluctuation
magnitude, indicating soil temperature at 5 cm started warming. For the CPLD model, CHF exceeds HC
during daytime and the difference increased with time. Negative values were found for LHF and developed
further over time. The CHF and LHF terms summed to nearly balance the HC term. Such behavior was

similarly reproduced by CPLD-AIR model with a slightly large difference between HC and CHF terms. This
means larger amount of water vapor was evaporated from 5 cm soil layer (with more negative LHF term)
from CPLD-AIR simulations than that from CPLD simulations, which explains the lower liquid water content
for CPLD-AIR model (Fig. 2, 5 cm).

### 3.5 Subsurface latent heat flux density

To give more context to the results, the spatial and temporal distribution of model simulated latent heat flux
density ($S_h$), $-\rho_w L \partial q_v / \partial z$, during the freezing and freezing-thawing transition period was shown as Fig. 7.
For the unCPLD model, the latent heat flux density ($S_h$) is not available due to its inability to depict the vapor
flow process.

Figure 7a shows that there is a strong diurnal variation of $S_h$ at upper 0.1cm soil layers. Such diurnal behavior

along the soil profile was interrupted by soil layer of 1cm, at which the water vapor consistently moved
upwards as evaporation source (termed as evaporative front). The path of this upward water vapor ended at
soil depth of 20cm from the 6[th] of December, where the freezing front developed. Compared to the upper
0.1cm soil, a weaker diurnal fluctuations of $S_h$ was found at lower soil layers.  For CPLD-AIR model, the
vapor transfer patterns are similar to that of CPLD model (Fig. 6b). There were isolated connections of

condensed water vapor between upper 1cm soil and the lower soil layers ($S_h>0$, e.g., 6[th], 7[th], 9[th], and 10[th] of
December), possibly associated with the downward air flow (see Fig. 12 in Yu et al. (2018)). The large
difference in magnitude of latent heat flux density between CPLD and CPLD-AIR model appeared mainly
isolated at upper soil layers (Fig. 7c). At soil layers between 1cm and 20cm, CPLD-AIR model simulated
less in condensation vapor area ($S_h>0$) and more in the evaporation area ($S_h<0$), indicating that CPLD-AIR


model produced an additional amount of condensation and evaporation water vapor compared with CPLD model (Fig. 7c).

Similar to that during the freezing period, strong diurnal variations at upper soil layers, interruption of diurnal patterns by the constant upward evaporation of intermediate soil layers, and weak diurnal variation at lower soil layers of $S_h$ can be clearly observed along soil profile during the freezing-thawing transition period (Fig.

7d, e). While the maximum evaporation rate was less than that during the freezing period. The consistent evaporation zone developed to a depth of 5 cm. The path for the upwards water vapor tended to develop deeper than 30cm with the absence of soil ice. The simulation by CPLD-AIR model produced more condensation and less evaporation water vapor than that by CPLD model can be seen more clearly (Fig. 7f). In addition, steadily more evaporation water vapor from soil depth of 5 cm was simulated by CPLD-AIR

model compared to CPLD model. This confirms the aforementioned point that during the freezing-thawing transition period, large LHF values were simulated by CPLD-AIR model (Fig. 6).

## 4. Discussion

### 4.1 Coupled Water and Heat Transfer Processes

The coupled water and heat transfer is realized via considering the vapor transfer processes. The mutual

dependence of soil water, in different phases (liquid, water vapor, and ice), and heat transport is enabled to facilitate our better understanding of the complex soil physical processes (e.g., Fig. 6-7). Specifically, the coevolution of soil moisture and soil temperature (SMST) profiles simulated by CPLD model was closer to the observation than that by unCPLD model. In addition, significant enhancement in portraying the monthly average diurnal variations of surface evapotranspiration and cumulative evapotranspiration can be found

from CPLD model simulations, which constraints the hydrothermal regimes especially during the freezing-thawing transition periods (Fig. 1, 2& 5). During the freezing period, liquid water in the soil freezes, which is analog to the soil drying process, and water vapor fluxes instead of liquid fluxes dominate the mass transfer process. Neglecting such important water flux component unavoidably results in unrealistic simulations of surface evapotranspiration and SMST profiles. From the energy budget perspective, the contribution of vapor

fluxes to the heat balance budget is more evidenced at soil layers above the evaporative front than that below it (e.g., Fig. 6a vs. Fig. 6d, corresponding evaporative front shown as Fig. 7a vs. Fig. 7d). The downward latent heat flux from CPLD model makes the subsurface soil warmer, which reduces the temperature gradient ($\partial T/\partial z$). This further results in the weaker diurnal fluctuations of conduction term for CPLD model than that for the unCPLD model (Fig. 6). At the soil layers below the evaporative front, the heat flux source from

vapor diffusion process (LHF) is negligible. Thermal retard effect as the presence of soil ice, expressed as the apparent heat capacity term ($C_{app}$), dominates the heat transfer process in frozen soils. CPLD model, considering the thermal effect on water flow, usually has a larger water capacity value $\partial\theta/\partial\psi$ than unCPLD model. As such, the intense thermal impedance effect leads to the results that CPLD model has a weaker diurnal fluctuation of soil temperature than unCPLD model at subsurface soil layers.



## 4.2 Air Flow in the Soil

Since soil pores are filled with liquid water, vapor and dry air, taking dry air as an independent state variable can facilitate better understanding of the relative contribution of each component in soil pores to the mass and heat transfer in soils. The results show that the dry air-induced water and heat flow is negligible to the total mass and energy transfer (Yu et al., 2018;Zeng et al., 2011a). Nevertheless, dry air can affect soil hydrothermal regimes significantly under certain circumstances. Wicky and Hauck (2017) reported that the airflow-induced convective heat transfer resulted in a considerable temperature difference between the upper and lower part of a permafrost talus slope and thus have a remarkable effect on the thermal regime of the talus slope. Zeng et al. (2011a) demonstrated the airflow-induced surface evaporation enhanced after precipitation events, since the hydraulic conductivity of topsoil layers increased tremendously due to the airflow from the atmosphere into the soil. In this study, we found that the explicit consideration of airflow have the model produce an additional amount of subsurface condensation and evaporative water vapor in the condensation region and evaporation region, respectively (Fig. 7c & f). The effect of latent heat flux on heat transfer was enhanced by airflow during the freezing-thawing transition period (Fig. 6), which further affects the subsurface hydrothermal simulations (e.g., Fig. 2).

## 5. Conclusions

On the basis of STMMUS modeling framework with various complexity of water and heat transfer physics (unCPLD, CPLD and CPLD-AIR model), the performance of each model in simulating water and heat transfer and surface evapotranspiration was tested on a typical Tibetan meadow. Results indicate that compared to the in situ observations, the unCPLD model tended to present an earlier freezing and thawing date with a stronger diurnal variation of soil temperature/liquid water in response to the atmospheric forcing. Such discrepancies were considerably reduced by model with the coupled water-heat physics. Surface evapotranspiration was overestimated by unCPLD model, mainly due to the mismatches during the initial freezing and freezing-thawing transition period. CPLD models, with the coupled constraints from the perspective of water and energy conservation, significantly improve the model performance in mimicking the surface evapotranspiration dynamics during frozen period. The analysis of heat budget components and latent heat flux density revealed that the improvement of soil temperature simulations by CPLD model is ascribed to its physical consideration of vapor flow and thermal effect on water flow, with the former mainly functions at regions above the evaporative front, the latter dominates below the evaporative front. The contribution of airflow induced water and heat flow to the total mass and energy fluxes is negligible. However, given the explicit consideration of airflow, the latent heat flux and its effect on heat transfer were enhanced during the freezing-thawing transition period. This work highlighted the role of considering the vapor flow, thermal effect on water flow, and airflow in portraying the subsurface soil hydrothermal dynamics especially during freezing-thawing transition periods. To sum up, this study can contribute to a better understanding of freeze-thaw mechanisms of permafrost, which will subsequently contribute to the quantification of



permafrost carbon feedback (Burke et al., 2013; Schaefer et al., 2014; Schuur et al., 2015), if the STEMMUS-

FT model is to be coupled with a biogeochemical model, as lately implemented (Yu et al., 2020).

*Data availability.* The soil hydraulic/thermal property data can be freely downloaded from 4TU. Center for
Research Data (https://doi.org/10.4121/uuid:61db65b1-b2aa-4ada-b41e-61ef70e57e4a). Some relevant data
are made available from 4TU. Center for Research Data (https://doi.org/10.4121/uuid:cc69b7f2-2448-
4379-b638-09327012ce9b).

*Author contribution.* Conceptualization, Z.S. and Y.Z.; methodology, L.Y., Y.Z.; writing - original draft
preparation, L.Y., Y.Z.; writing – review & editing, L.Y.,Y.Z., Z.S..


*Competing interests.* The authors declare that they have no conflict of interest.

**Acknowledgment**

This work is supported by the National Natural Science Foundation of China (grant no. 41971033) and
supported by the Fundamental Research Funds for the Central Universities, CHD (grant no. 300102298307).
The authors thank the editor and referees very much for their constructive comments on improving the
manuscript.


**Notation**

| Parameter | Symbol | Unit | Value |
|---|---|---|---|
| Volumetric water content | $\theta$ | m³ m⁻³ | |
| Water flux | $q$ | kg m⁻² s⁻¹ | |
| Vertical space coordinate (positive upwards) | $z$ | m | |
| Sink term for transpiration, evaporation | $S$ | s⁻¹ | |
| Density of soil liquid water | $\rho_L$ | kg m⁻³ | 1000 |
| Hydraulic conductivity | $K$ | m s⁻¹ | |
| Water potential | $\psi$ | m | |
| Time | $t$ | s | |
| Heat capacity of the bulk soil | $C_{soil}$ | J kg⁻¹ °C⁻¹ | |
| Soil temperature | $T$ | °C | |
| Effective thermal conductivity of the soil | $\lambda_{eff}$ | W m⁻¹ °C⁻¹ | |
| Latent heat of fusion | $L_f$ | J kg⁻¹ | 3.34E5 |
| Soil ice volumetric water content | $\theta_I$ | m³ m⁻³ | |
| Density of water vapor | $\rho_V$ | kg m⁻³ | |
| Density of ice | $\rho_i$ | kg m⁻³ | 920 |
| Soil liquid volumetric water content | $\theta_L$ | m³ m⁻³ | |
| Soil vapor volumetric water content | $\theta_V$ | m³ m⁻³ | |
| Soil liquid water fluxes (positive upwards) | $q_L$ | kg m⁻² s⁻¹ | |
| Soil water vapor fluxes (positive upwards) | $q_V$ | kg m⁻² s⁻¹ | |
| Isothermal hydraulic conductivities | $K_{Lh}$ | m s⁻¹ | |
| Thermal hydraulic conductivities | $K_{LT}$ | m² s⁻¹ °C⁻¹ | |
| Isothermal vapor conductivity | $D_{Vh}$ | kg m⁻² s⁻¹ | |
| Thermal vapor diffusion coefficient | $D_{VT}$ | kg m⁻¹ s⁻¹ °C⁻¹ | |
| Specific heat capacity of soil solids | $C_s$ | J kg⁻¹ °C⁻¹ | |
| Specific heat capacity of liquid | $C_L$ | J kg⁻¹ °C⁻¹ | 4.186 |
| Specific heat capacity of water vapor | $C_V$ | J kg⁻¹ °C⁻¹ | 1.87 |
| Specific heat capacity of ice | $C_i$ | J kg⁻¹ °C⁻¹ | 2.0455 |
| Density of solids | $\rho_s$ | kg m⁻³ | |
| Volumetric fraction of solids in the soil | $\theta_s$ | m³ m⁻³ | |
| Arbitrary reference temperature | $T_r$ | °C | 20 |





| | | | |
|---|---|---|---|
| Latent heat of vaporization of water at the reference temperature | $L_0$ | J kg$^{-1}$ | |
| Differential heat of wetting | $W$ | J kg$^{-1}$ | |
| Liquid water flux driven by the gradient of matric potential | $q_{Lh}$ | kg m$^{-2}$ s$^{-1}$ | |
| Liquid water flux driven by the gradient of matric potential | $q_{LT}$ | kg m$^{-2}$ s$^{-1}$ | |
| Liquid water flux driven by the gradient of air pressure | $q_{La}$ | kg m$^{-2}$ s$^{-1}$ | |
| Water vapor flux driven by the gradient of matric potential | $q_{Vh}$ | kg m$^{-2}$ s$^{-1}$ | |
| Water vapor flux driven by the gradient of matric potential | $q_{VT}$ | kg m$^{-2}$ s$^{-1}$ | |
| Water vapor flux driven by the gradient of air pressure | $q_{Va}$ | kg m$^{-2}$ s$^{-1}$ | |
| Mixed pore-air pressure | $P_g$ | Pa | |
| Specific weight of water | $\gamma_W$ | kg m$^{-2}$ s$^{-2}$ | |
| Transport coefficient for adsorbed liquid flow due to temperature gradient | $D_{TD}$ | kg m$^{-1}$ s$^{-1}$ °C$^{-1}$ | |
| Isothermal vapor conductivity | $D_{Vh}$ | kg m$^{-2}$ s$^{-1}$ | |
| Thermal vapor diffusion coefficient | $D_{VT}$ | kg m$^{-1}$ s$^{-1}$ °C$^{-1}$ | |
| Advective vapor transfer coefficient | $D_{Va}$ | s | |
| Specific heat capacity of dry air | $C_a$ | J kg$^{-1}$ °C$^{-1}$ | 1.005 |
| Liquid water flux | $q_L$ | kg m$^{-2}$ s$^{-1}$ | |
| Vapor water flux | $q_V$ | kg m$^{-2}$ s$^{-1}$ | |
| Dry air flux | $q_a$ | kg m$^{-2}$ s$^{-1}$ | |
| Porosity | $\varepsilon$ | - | |
| Density of dry air | $\rho_{da}$ | kg m$^{-3}$ | |
| Degree of saturation in the soil | $S_L$ | - | $=\theta_L/\varepsilon$ |
| Degree of air saturation in the soil | $S_a$ | - | $=1-S_L$ |
| Henry's constant | $H_c$ | - | 0.02 |
| Molecular diffusivity of water vapor in soil | $D_e$ | m$^2$ s$^{-1}$ | |
| Intrinsic air permeability | $K_g$ | m$^2$ | |
| Air viscosity | $\mu_a$ | kg m$^{-2}$ s$^{-1}$ | |
| Volumetric fraction of dry air in the soil | $\theta_a$ | m$^3$ m$^{-3}$ | $=\theta_V$ |
| Gas phase longitudinal dispersion coefficient | $D_{Vg}$ | m$^2$ s$^{-1}$ | |
| Soil saturated hydraulic conductivity | $K_s$ | m s$^{-1}$ | |
| Saturated soil water content | $\theta_s$ | m$^3$ m$^{-3}$ | |
| Residual soil water content | $\theta_r$ | m$^3$ m$^{-3}$ | |
| Air entry value of soil | $\alpha$ | m$^{-1}$ | |





| | | | |
|---|---|---|---|
| Van Genuchten fitting parameters | $n$ | - | |
| Apparent heat capacity | $C_{app}$ | J kg$^{-1}$ °C$^{-1}$ | $= C_{soil} + \rho_i \dfrac{L_f^2}{gT}\dfrac{d\theta}{d\psi}$ |
| Latent heat flux density | $S_h$ | W m$^{-3}$ | $= -\rho_w L \partial q_v / \partial z$ |




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





**Tables and Figures**

**Table 1. Governing equations for different complexity of water and heat coupling physics (See appendix for notations)**

| Models | Governing equations (water, heat and air) | Number |
|---|---|---|
| unCPLD | $\frac{\partial \theta}{\partial t} = -\frac{\partial q}{\partial z} - S = \rho_L \frac{\partial}{\partial z}\left[K\left(\frac{\partial \psi}{\partial z} + 1\right)\right] - S$ | (1) |
| | $C_{soil}\frac{\partial T}{\partial t} - \rho_i L_f \frac{\partial \theta_i}{\partial t} = \frac{\partial}{\partial z}\left(\lambda_{eff}\frac{\partial T}{\partial z}\right)$ | (2) |
| CPLD | $\frac{\partial}{\partial t}(\rho_L\theta_L + \rho_V\theta_V + \rho_i\theta_i) = -\frac{\partial}{\partial z}(q_L + q_V) - S$ | |
| | $= -\frac{\partial}{\partial z}(q_{Lh} + q_{LT} + q_{Vh} + q_{VT}) - S$ | (3) |
| | $= \rho_L\frac{\partial}{\partial z}\left[K_{Lh}\left(\frac{\partial \psi}{\partial z} + 1\right) + K_{LT}\frac{\partial T}{\partial z}\right] + \frac{\partial}{\partial z}\left[D_{Vh}\frac{\partial \psi}{\partial z} + D_{VT}\frac{\partial T}{\partial z}\right] - S$ | |
| | $\frac{\partial}{\partial t}\left[(\rho_s\theta_s C_s + \rho_L\theta_L C_L + \rho_V\theta_V C_V + \rho_i\theta_i C_i)(T - T_r) + \rho_V\theta_V L_0 - \rho_i\theta_i L_f\right] - \rho_L W \frac{\partial \theta_L}{\partial t}$ | (4) |
| | $= \frac{\partial}{\partial z}\left(\lambda_{eff}\frac{\partial T}{\partial z}\right) - \frac{\partial}{\partial z}\left[q_L C_L(T - T_r) + q_V(L_0 + C_V(T - T_r))\right] - C_L S(T - T_r)$ | |
| CPLD-AIR | $\frac{\partial}{\partial t}(\rho_L\theta_L + \rho_V\theta_V + \rho_i\theta_{ice}) = -\frac{\partial}{\partial z}(q_{Lh} + q_{LT} + q_{La} + q_{Vh} + q_{VT} + q_{Va}) - S$ | |
| | $= \rho_L\frac{\partial}{\partial z}\left[K\left(\frac{\partial \psi}{\partial z} + 1\right) + D_{TD}\frac{\partial T}{\partial z} + \frac{K}{\gamma_w}\frac{\partial P_g}{\partial z}\right] + \frac{\partial}{\partial z}\left[D_{Vh}\frac{\partial \psi}{\partial z} + D_{VT}\frac{\partial T}{\partial z} + D_{Va}\frac{\partial P_g}{\partial z}\right]$ | (5) |
| | $\quad - S$ | |
| | $\frac{\partial}{\partial t}\left[(\rho_s\theta_s C_s + \rho_L\theta_L C_L + \rho_V\theta_V C_V + \rho_{da}\theta_a C_a + \rho_i\theta_i C_i)(T - T_r) + \rho_V\theta_V L_0 - \rho_i\theta_i L_f\right] - \rho_L W \frac{\partial \theta_L}{\partial t}$ | (6) |
| | $= \frac{\partial}{\partial z}\left(\lambda_{eff}\frac{\partial T}{\partial z}\right) - \frac{\partial}{\partial z}\left[q_L C_L(T - T_r) + q_V(L_0 + C_V(T - T_r)) + q_a C_a(T - T_r)\right]$ | |
| | $\quad - C_L S(T - T_r)$ | |
| | $\frac{\partial}{\partial t}[\varepsilon\rho_{da}(S_a + H_c S_L)]$ | |
| | $\qquad = \frac{\partial}{\partial z}\left[D_e\frac{\partial \rho_{da}}{\partial z} + \rho_{da}\frac{S_a K_g}{\mu_a}\frac{\partial P_g}{\partial z} - H_c\rho_{da}\frac{q_L}{\rho_L} + \left(\theta_a D_{Vg}\right)\frac{\partial \rho_{da}}{\partial z}\right]$ | (7) |





**Table 2. The adopted average values of soil texture and hydraulic properties at different depths (See appendix for notations)**

| Soil depth (cm) | Clay (%) | Sand (%) | $K_s$ ($10^{-6}$ m s$^{-1}$) | $\theta_s$ (m$^3$ m$^{-3}$) | $\theta_r$ (m$^3$ m$^{-3}$) | $\alpha$ (m$^{-1}$) | $n$ |
|---|---|---|---|---|---|---|---|
| 5-10 | 9.00 | 44.13 | 1.45 | 0.50 | 0.035 | 0.041 | 1.332 |
| 10-40 | 10.12 | 44.27 | 0.94 | 0.45 | 0.039 | 0.041 | 1.362 |
| 40-160 | 5.59 | 65.55 | 0.68 | 0.41 | 0.045 | 0.075 | 1.590 |

435

**Table 3. Comparative statistics values of observed and simulated soil temperature/moisture with three models, with the bold fonts indicating the best statistical performance**

| Experiment | Statistics | Soil temperature (ºC) | | | | | Soil moisture (m$^3$ m$^{-3}$) | | | | |
|---|---|---|---|---|---|---|---|---|---|---|---|
| | | 5cm | 10cm | 20cm | 40cm | 80cm | 5cm | 10cm | 20cm | 40cm | 80cm |
| unCPLD | BIAS | **-0.039** | 0.177 | -0.022 | -1.103 | -0.140 | 0.009 | 0.009 | 0.005 | **0.004** | 0.002 |
| | RMSE | 0.381 | 0.407 | 0.521 | 1.524 | 0.526 | 0.025 | 0.022 | 0.031 | 0.032 | 0.012 |
| CPLD | BIAS | -0.183 | 0.093 | **0.001** | -0.956 | **0.027** | **0.000** | 0.004 | 0.001 | 0.005 | **0.001** |
| | RMSE | 0.365 | **0.314** | 0.186 | 1.168 | 0.128 | **0.008** | 0.007 | 0.003 | 0.007 | **0.002** |
| CPLD-AIR | BIAS | -0.187 | **0.093** | 0.005 | **-0.953** | 0.029 | -0.001 | **0.004** | **0.001** | 0.005 | 0.001 |
| | RMSE | **0.362** | 0.316 | **0.180** | 1.168 | **0.126** | 0.011 | **0.006** | **0.003** | **0.007** | 0.002 |

Note: $BIAS = \frac{\sum_{i=1}^{n}(y_i - \hat{y}_i)}{n}$, $RMSE = \sqrt{\frac{\sum_{i=1}^{n}(y_i - \hat{y}_i)^2}{n}}$, where $y_i$, $\hat{y}_i$ are the measured and model simulated soil

440    temperature/moisture; n is the number of data points.

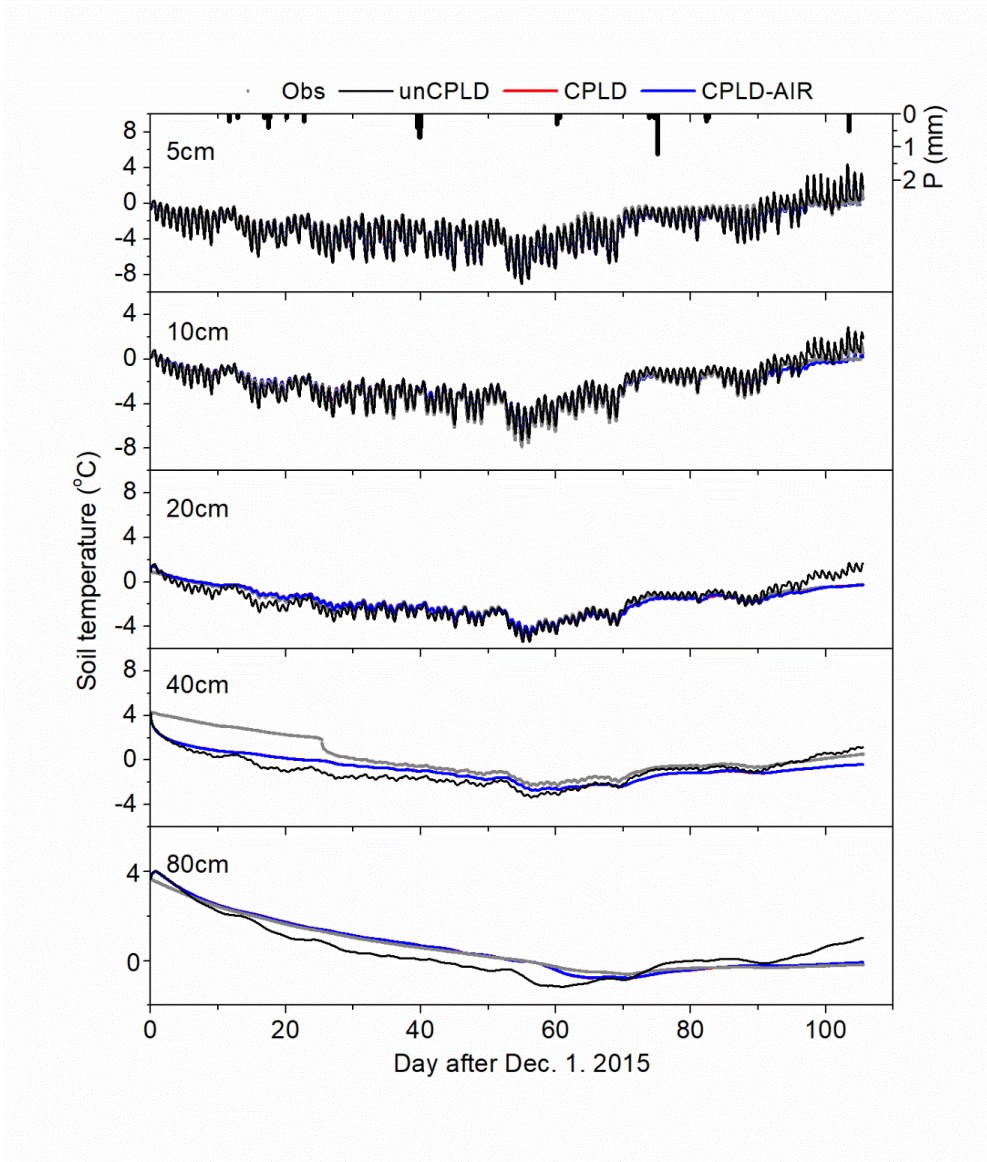

**Figure 1. Comparison of measured (Obs) and estimated time series of soil temperature at various soil layers using uncoupled soil physics (unCPLD), coupled water and heat physics (CPLD) and coupled water and heat physics with air flow (CPLD-AIR) model.**

445

**Figure 2. Comparison of measured (Obs) and model simulated time series of soil moisture at various soil layers using uncoupled soil physics (unCPLD), coupled water and heat physics (CPLD) and coupled water and heat physics with air flow (CPLD-AIR) model.**

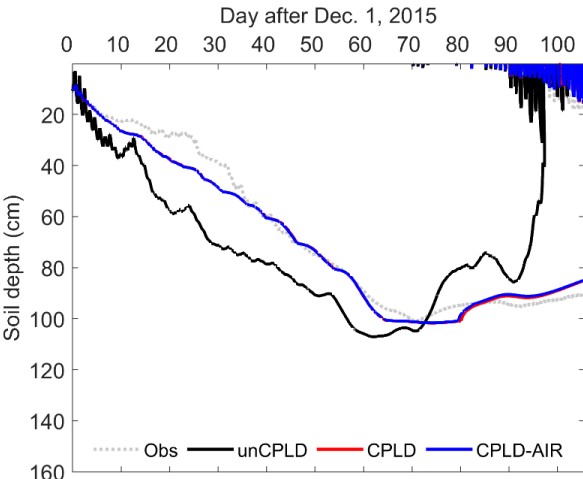

450    **Figure 3. Comparison of measured (Obs) and model simulated freezing front propagation (FFP) using uncoupled soil physics (unCPLD), coupled water and heat physics (CPLD) and coupled water and heat physics with air flow (CPLD-AIR) model. Note the measured FFP was seen as the development of zero degree isothermal lines from the measured soil temperature field.**



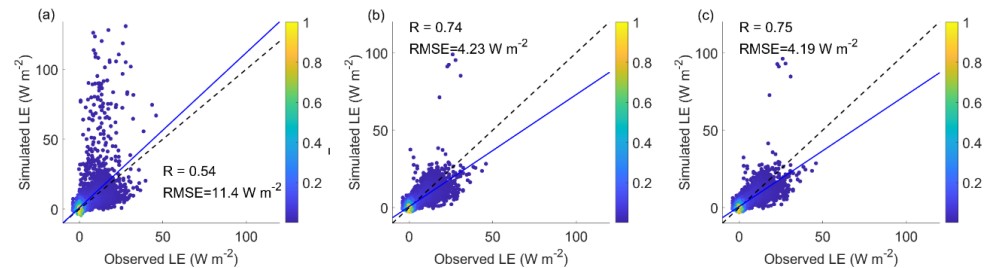

**Figure 4. Scatter plot of observed and model estimated half-hourly latent heat flux using (a) uncoupled soil physics (unCPLD), (b) coupled water and heat physics (CPLD) and (c) coupled water and heat physics with air flow (CPLD-AIR) model. The color indicates the data composite of surface latent heat flux.**

455





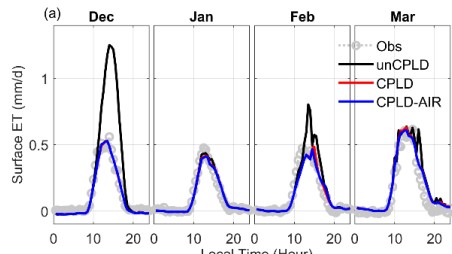
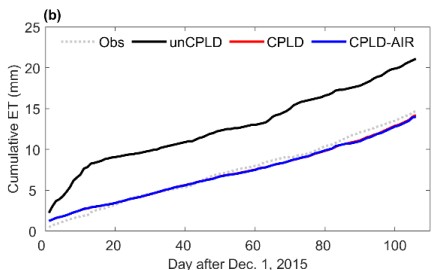

**Figure 5. Comparison of observed and model simulated (a) mean diurnal variations of surface evapotranspiration and (b) cumulative evapotranspiration (ET) by unCPLD, CPLD, and CPLD-AIR model.**



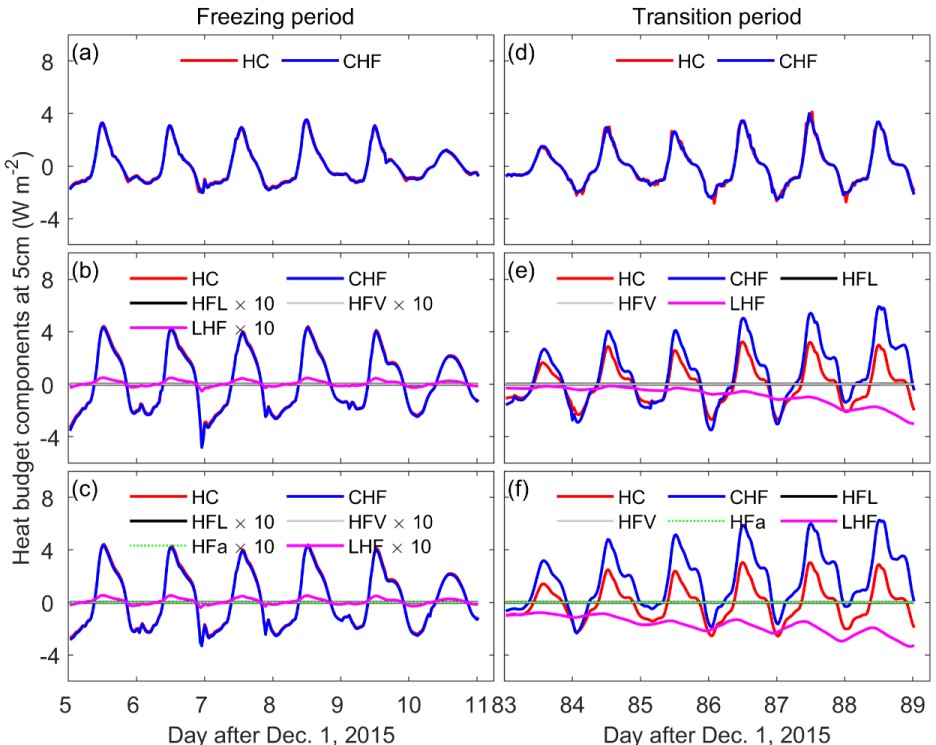

Figure 6. Time series of model simulated heat budget components at the soil depth of 5cm using (a &d) unCPLD, (b &e) CPLD, and (c &f) CPLD-AIR simulations during the typical 6-day freezing (left column) and freezing-thawing transition (right column) periods. HC, rate of change of heat content, CHF, conductive heat flux divergence, HFL, convective heat flux divergence due to liquid water flow, HFV, convective heat flux divergence due to water vapor flow, HFa, convective heat flux divergence due to air flow, LHF, latent heat flux divergence. Note that for graphical purposes, HFL, HFV, HFa, and LHF were enhanced by a factor of 10 during the freezing period.

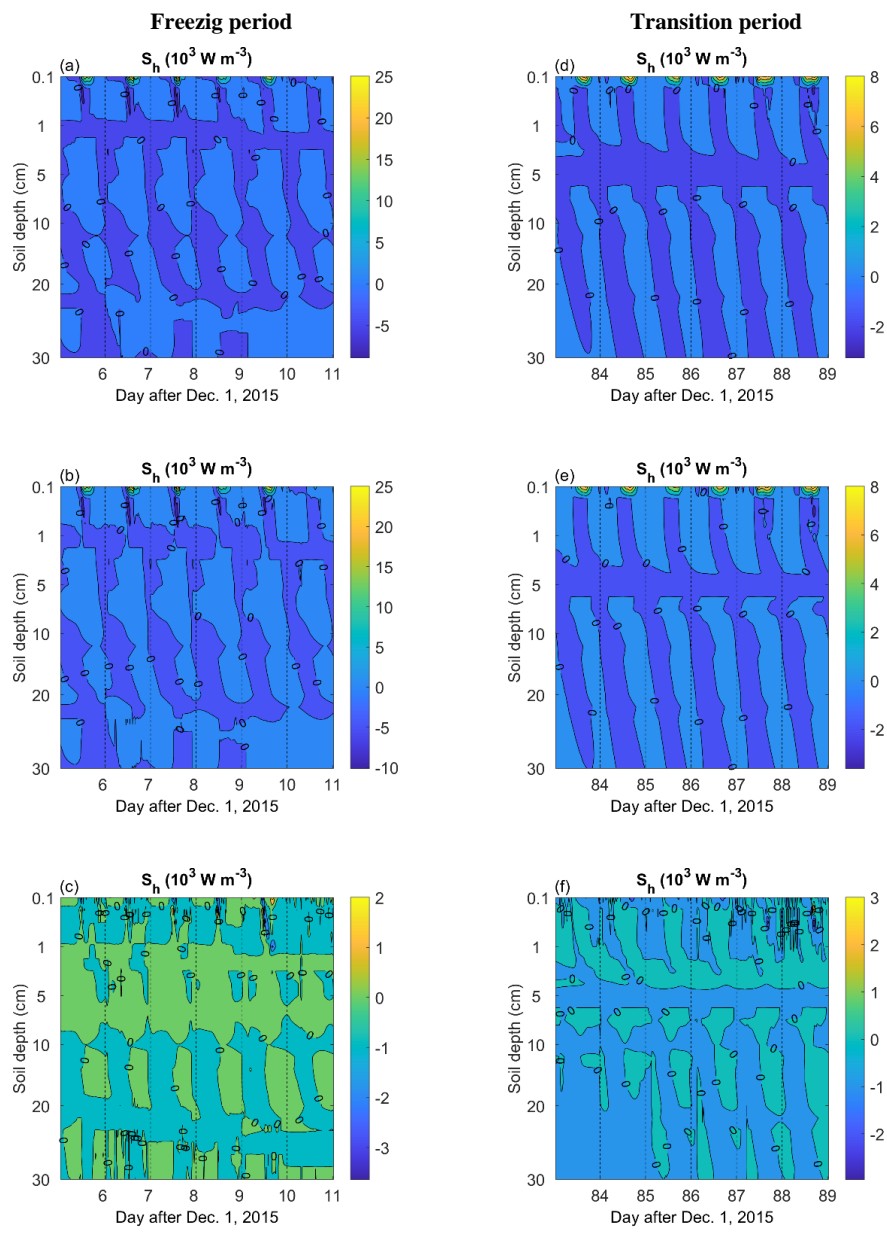

**Figure 7. The spatial and temporal distributions of model estimated soil latent heat flux density using (a &d) CPLD, (b &e) CPLD-AIR and (c &f) the difference between CPLD and CPLD-AIR simulations ($S_{h,CPLD-AIR} - S_{h,CPLD}$) during the typical 6-day freezing and freezing-thawing transition periods. The left and right column are for the freezing and freezing-thawing transition period, respectively. Note that figures for the unCPLD model are absent as it can not simulate the subsurface soil latent heat flux density.**

470