# Peer review of "Understanding the Mass, Momentum and Energy Transfer in the Frozen Soil with Three Levels of Model Complexities"

_Hydrology and Earth System Sciences, 2020_

## Referee Comment (RC1) · Orgogozo Laurent (Referee) · 6 Jul 2020

General comments

The considered work deals with the physics of the heat and water transfers in seasonally frozen soils, and in particular with the importance of the descriptions of the couplings that occur in these transport phenomena, from the basic couplings due freeze/thaw of the pore water (latent heat of solidification/liquefaction, impact of freezing on the hydraulic properties) to finer effects such as those related to heat gradient induced water flow or to the water vapor fluxes, and even to the effect of dry air flow. Thermo-hydrological transfers modeling in seasonally frozen soils is a complex prob-
lem with various important implications as it is well explained in the introduction section, and the handling of couplings is one of the major difficulties for their numerical simulations, thus I feel that the scope of the manuscript is appropriate for a submission to HESS.

The authors propose a comparative analysis of numerical simulations of ground thermo-hydrological status in a mountain frozen soil field site for which observations are available for a winter season. After a brief description of the considered field site and of the numerical models to be used, the obtained numerical results are presented. Finally the comparison of results obtained with various physical assumptions embedding various level of complexity of the multi-physics couplings involved allows the authors to make a discussion on the trade-off that must be made between the accuracy of the simulations and the complexity of the modeling approach.

The goal of the work and its real interest for the study of cold regions hydrology are clearly described, while the proposed methodology is relevant for such a purpose. Nevertheless some critical information are missing in the descriptions of the equations and of the numerical procedures, which damages the completeness of the manuscript, and prevents the reader to assess the range of validity of the conclusions. Moreover the domain of applicability of these conclusions in terms of biogeoclimatic contexts should be better discussed. Thus I suggest a major revision of this manuscript prior to publication. One can find below the specific comments on which are based the previous statement, along with a few technical corrections.

Specific comments

Abstract:

l22 : 'air-flow induced water (...) transport (...) is negligible': what is the difference with vapor flow, that have been stated as significant in the previous sentence ? Please clarify.

2. Methodology 2.2 Mass and energy transport in unsaturated soils

l105-109 : The latent heat due to water freeze/thaw introduces necessarily a coupling between heat transport and water transport, since the latent heat term in the thermal equation depends on the water content of the porous medium. The effect of soil ice on soil hydraulic properties induces also a coupling between heat transport and water transport, since the hydraulic properties then depend on the temperature of the porous medium (at least whether the temperature is above or below $0°C$). Thus the name 'uncoupled' is inappropriate for describing the set of equation in the most simple model ('unCPLD' model), and its use is not in line with the common practices in the field of cryohydrogeological modeling (e.g.: Grenier et al., 2018). In fact both 'unCPLD' model and 'CPLD' model are coupled thermo-hydrological models, but the latter embeds more coupling effects than the former. Basic coupled model (BCM) versus Advanced coupled model (ACM) might be a better terminology for instance ?

C. Grenier, H. Anbergen, V. Bense, Q. Chanzy, E. Coon, N. Collier, F. Costard, M. Ferry, A. Frampton, J. Frederick, J. Gonçalvès, J. Holmén, A. Jost, S. Kokh, B. Kurylyk, J. McKenzie, J. Molson, E. Mouche, L. Orgogozo, R. Pannetier, A. Rivière, N. Roux, W. Rühaak, J. Scheidegger, J.-O. Selroos, R. Therrien, P. Vidstrand, C. Voss, 2018. Groundwater flow and heat transport for systems undergoing freeze-thaw: Intercomparison of numerical simulators for 2D test cases. Adv. Water Resour., 114, 196-218.

l103 – section 2.2 : a clear presentation of the boundary conditions used for each considered equations in each models is missing. As they are important information for the understanding of the numerical results, they should be added. Numerical convergence studies (meshes resolutions, used time steps, ...) must also be evocated here: in order to compare the results of different models, it is important to control that the truncation errors are comparable between each models (and small compared to the discussed inter-model discrepancies!).

3. Results

l172 and following : The numerical results in terms of computed evapotranspiration depend critically on the way to parameterize evapotranspiration, which is not presented in the paper. Various descriptions could be used here. For instance, and among many others, an empirical one emphasizing the role of vegetation could be find in Orgogozo et al., 2019, or a theoretically derived one in the case of purely evaporative processes could be find in Duval et al., 2004. The mathematical expressions and input data used to compute the evapotranspiration in each model should be described in the manuscript. Without these key information, it is not possible for the reader to interpret the given results.

F. Duval, F. Fichot, M. Quintard, 2004. A local thermal non-equilibrium model for two-phase flows with phase-change in porous media. International Journal of Heat and Mass Transfer 47 : 613-639.

L. Orgogozo, A.S. Prokushkin, O.S. Pokrovsky, C. Grenier, M. Quintard, J. Viers, S. Audry, 2019. Water and energy transfer modelling in a permafrost-dominated, forested catchment of Central Siberia: the key role of rooting depth. Permafrost and Periglacial Processes 30 : 75-89.

4. Discussion

l244 : The first sentence is wrong: the vapor transfer processes are not the only sources of couplings between thermal and hydrological transfers in porous media when freeze/thaw of the pore water occurs, see also my first comment on section 2. Methodology.

ll254-l264 : Here is the explanation for the difference of amplitude of diurnal cycle between models. It seems to me that this is the key point of the discussion (evocated already numerous times in the manuscript, e.g;: l141, l147, l163), but somewhat hard to follow. It should be rewritten in a clearer way, may be with explicative schemes ?

l274 : 'hydraulic conductivity' increase due to 'airflow from the atmosphere to the soil'
? Please give a short explanation.

5. Conclusion

l286-290 : The strong point made about evapotranspiration highlights the need to give to the reader all the relevant information about the handling of the evapotranspiration sink terms in each model (see also my comment for the section 3. Results). Please discuss also the transpirative component of evapotranspiration.

l296-301 : The domain of applicability of the presented study should be better discussed. For instance, a point is made about the freeze/thaw mechanisms of permafrost while the studied field site is not in a permafrost affected area. The relative importance of the vapor flow, the thermal effect on water flow and the airflow should be more discussed with respect to the biogeoclimatic context (e.g. : more important in climate with long freezing/thawing periods or with long periods with surface temperature oscillating around 0°C), and in the context of the existing literature (e.g.: Karra et al., 2014).

S. Karra, S.L. Painter, P.C. Lichtner, 2014. Three-phase numerical model for subsurface hydrology in permafrost-affected regions (PFLOTRAN-ICE v1.0). The Cryosphere 8(5) : 1935-1950.

Technical comments :

The English language should be improved, although I am not a native English-speaking person so maybe I am making a mistake on that point. For instance it seems to me that the vegetation development cannot be 'closed' (l29). As another example I think that 'the best water resources management' or 'a better water resources management' could be used but not 'the better water resources management' (l33). A reread by an English editing service might be helpful.

l40 and also in other places (e.g.: l42, l51) : Citations should be re-ordered (2006 before 2010).

l138 : Fig1. The figure is not clear enough. Firstly it is difficult to decipher the different

curves for 5, 10 and 20 cm depth – CPLD curve is nowhere visible (if it si beneath the CPLD-Air curve, make this one discontinuous). The legend should also be clearer for the obs. Secondly I didn't got the 'earlier stepping in / stepping out of the frozen period', may be they should be pointed out in the figure itself.

l147 : Fig2. Same formal remarks that for Fig1.

---

## Referee Comment (RC2) · Anonymous Referee #2 · 13 Jul 2020

Frozen soil undergoing freeze-thaw cycles has significant impacts on local hydrology, ecosystems, and engineering infrastructure within the context of global warming. However, it is challenging to depict a dynamic thermal equilibrium system of ice, liquid water, water vapor and dry air in soil pores, when soil experiences the freeze/thaw process. Through careful design and analyses of numerical simulation experiments, this study may help us understand the contribution of airflow-induced water and heat transport in the frozen ground. I just have a few comments/suggestions that may improve this manuscript, before it can be accepted for publication in HESS.

**Major comments:**

1. The authors should clearly state/add the innovative points by this study in the title, abstract, and body text (e.g., objectives, results and discussions, as well as conclusions), by comparing to the listed publications in the references by the same authors. It is obvious that this group has quite a few nice publications on the physics of frozen ground, by describing the contributions/roles of vapor, liquid water and solid ice in the water and heat transports. After my reading of this manuscript, it is more like a sensitive study or a review paper. Please add text to clarify the major difference of this manuscript from previous studies, and demonstrate the new processes/knowledge to the permafrost hydrology community.

2. In Figure (1-3 & 5), the red and blue lines are always overlapped. Is there a better way to show them?

3. The difference between CLPD-air and CLP is that air flow was taken into account. What are the key processes that the air flow affects frozen ground? The difference should be briefly introduced in Section 2.2, for better understanding in this

manuscript.

4. There are many results in this paper, and I think you can add more details in Section 5 (conclusions).

5. Literature review about the frozen ground/permafrost hydrology by this manuscript is incomplete. I would like to suggest the authors also referring to the following ones but not limited to them. E.g.,

- *Qi et al. (2019). Coupled Snow and Frozen Ground Physics Improves Cold Region Hydrological Simulations: An Evaluation at the Upper Yangtze River Basin (Tibetan Plateau). Journal of Geophysical Research: Atmospheres, 124(33): 12985-13004.*

- *Biskaborn et al. (2019).Permafrost is warming at a global scale. Nature communications, 10(1), 264.*

- *Wang et al. (2017). Development of a land surface model with coupled snow and frozen soil physics. Water Resources Research, 53, 5085–5103.*

- *Bao et al. (2016). Development of an enthalpy‐based frozen soil model and its validation in a cold region in China. Journal of Geophysical Research: Atmospheres, 121(10), 5259-5280.*

- *Iijima, Y., Ohta, T., Kotani, A., Fedorov, A.N., Kodama, Y., & Maximov, T.C. (2014). Sap flow changes in relation to permafrost degradation under increasing precipitation in an eastern Siberian larch forest. Ecohydrology, 7(2), 177-187.*

---

## Author Response (AR1)

We appreciate very much the editor and reviewers on reading through this manuscript and posting useful comments. We presented the point by point response to the reviewers' comments. The comments are in black fonts and our responses are in blue fonts.

**Referee #1 (Orgogozo Laurent)**

General comments

The considered work deals with the physics of the heat and water transfers in seasonally frozen soils, and in particular with the importance of the descriptions of the couplings that occur in these transport phenomena, from the basic couplings due freeze/thaw of the pore water (latent heat of solidification/liquefaction, impact of freezing on the hydraulic properties) to finer effects such as those related to heat gradient induced water flow or to the water vapor fluxes, and even to the effect of dry air flow. Thermo-hydrological transfers modeling in seasonally frozen soils is a complex problem with various important implications as it is well explained in the introduction section, and the handling of couplings is one of the major difficulties for their numerical simulations, thus I feel that the scope of the manuscript is appropriate for a submission to HESS.

The authors propose a comparative analysis of numerical simulations of ground thermo-hydrological status in a mountain frozen soil field site for which observations are available for a winter season. After a brief description of the considered field site and of the numerical models to be used, the obtained numerical results are presented. Finally the comparison of results obtained with various physical assumptions embedding various level of complexity of the multi-physics couplings involved allows the authors to make a discussion on the trade-off that must be made between the accuracy of the simulations and the complexity of the modeling approach.

The goal of the work and its real interest for the study of cold regions hydrology are clearly described, while the proposed methodology is relevant for such a purpose. Nevertheless some critical information are missing in the descriptions of the equations and of the numerical procedures, which damages the completeness of the manuscript, and prevents the reader to assess the range of validity of the conclusions. Moreover the domain of applicability of these conclusions in terms of biogeoclimatic contexts should be better discussed. Thus I suggest a major revision of this manuscript prior to publication. One can find below the specific comments on which are based the previous statement, along with a few technical corrections.

**Response:** Thanks a lot for your constructive comments. We added the descriptions of the equations and the numerical procedures accordingly in Sect. 2.2. We added the Figure 1 to illustrate the boundary conditions, mesh resolutions, and half-hourly measurements of the driving force during our simulation period (the figure numbers were thus changed). The equations to calculate evapotranspiration were added in the Appendix. We briefly presented the relevant studies to corroborate our results in the Discussion part (Sect. 4.1). Please find our specific response as follows.

Specific comments

1. Abstract: l22 : 'air-flow induced water (...) transport (...) is negligible': what is the difference with vapor flow, that have been stated as significant in the previous sentence ? Please clarify.

**Response**: The air flow induced water and heat transport is refer to the water and vapor flow driven by the air pressure gradient, i.e., $\rho_L \frac{\partial}{\partial z}(\frac{K}{\gamma_w}\frac{\partial P_g}{\partial z})$ and $\frac{\partial}{\partial z}(D_{Va}\frac{\partial P_g}{\partial z})$ in Eq. 5 and its corresponding heat

flow $-\frac{\partial}{\partial z}(q_a C_a(T - T_r))$ in Eq. 6. Its contribution to the total energy transfer is negligible, as the term HFa shown in Figure 7.

Vapor flow is the isothermal and thermal vapor flow driven by the soil matric potential gradient and temperature gradient, i.e., $\frac{\partial}{\partial z}(D_{Vh}\frac{\partial \psi}{\partial z})$ and $\frac{\partial}{\partial z}(D_{VT}\frac{\partial T}{\partial z})$, respectively. The heat flux driven by the vapor flow refers to $-\frac{\partial}{\partial z}(q_V(L_0 + C_V(T - T_r))])$. From Figure 8, we can find that the relevant latent heat flux density Sh is significant at the upper soil layers, see also (LHF+HFV) term for the heat flow in Figure 7.

We rephrased the text as "The contribution of airflow-induced water and heat transport (driven by the air pressure gradient) to the total mass and energy fluxes is negligible."

2. Methodology 2.2 Mass and energy transport in unsaturated soils

l105-109 : The latent heat due to water freeze/thaw introduces necessarily a coupling between heat transport and water transport, since the latent heat term in the thermal equation depends on the water content of the porous medium. The effect of soil ice on soil hydraulic properties induces also a coupling between heat transport and water transport, since the hydraulic properties then depend on the temperature of the porous medium (at least whether the temperature is above or below 0_C). Thus the name 'uncoupled' is inappropriate for describing the set of equation in the most simple model ('unCPLD' model), and its use is not in line with the common practices in the field of cryohydrogeological modeling (e.g.: Grenier et al., 2018). In fact both 'unCPLD' model and 'CPLD' model are coupled thermo-hydrological models, but the latter embeds more coupling effects than the former. Basic coupled model (BCM) versus Advanced coupled model (ACM) might be a better terminology for instance ?

**Response**: Many thanks for pointing out it. For the simple 'unCPLD' model, there are indeed water and heat coupling mechanisms considered during the frozen period. The coupling between water and heat transport in 'unCPLD' model is achieved by the latent heat term due to phase change and the effect of soil ice on hydraulic properties.
The advanced CPLD model taken into account the water and heat coupling mechanisms during both the unfrozen and frozen periods. The vapor flow, which is the function of both soil moisture and temperature, makes the water and heat transfer tightly coupled. The thermal effect on soil matric potential and hydraulic conductivity, from the soil water surface tension and viscous flow effect, have the water flow dependent on the temperature. The convective heat flow in the energy conservation equation, which is due to the liquid/vapor fluxes, makes the heat transport dependent on the soil water flow.
We agree to use the suggested terminology as Basic coupled model (BCM), Advanced coupled model (ACM) and Advanced coupled model with air (ACM-AIR). The changes were made throughout this manuscript.

3. l103 – section 2.2 : a clear presentation of the boundary conditions used for each considered equations in each models is missing. As they are important information for the understanding of the numerical results, they should be added. Numerical convergence studies (meshes resolutions, used time steps, ...) must also be evocated here: in order to compare the results of different models, it is important to control that the truncation errors are comparable between each models (and small compared to the discussed inter-model discrepancies!).

**Response**: We added the description of the boundary conditions in the Sect. 2.2 as "Surface boundary for the water transport was set as the flux-type boundary controlled by the atmospheric forcing condition (i.e., evaporation, precipitation) while the specific soil temperature was assigned as the

surface boundary of energy conservation equation. The free drainage (zero matric potential gradient) and measured soil temperature were set as the bottom boundary conditions for the water transport and heat transport, respectively. For the air flow, the surface boundary was set as the atmospheric forcing condition and soil air was allowed to escape from the bottom of soil column."
We added Figure 1 to illustrate the model-used boundary condition, mesh resolutions, and driving forces.

The truncation errors due the numerical solution are related to the node distance and time steps. We added such description in Sect. 2.2 as "The vertical soil discretization was designed finer for the upper soil layers (0.1-2.5cm for 0-40cm, 27 layers) than that for the lower soil layers (5-20cm for 40-160cm, 10 layers). The adaptive time step strategy, with maximum time steps ranging from 1s to 1800s, was utilized for the numerical solution."
Note that all three models employed the same mesh resolutions and adaptive time step strategies. It indicated that the truncation errors due to numerical solution among three models are comparable. The difference is mainly restricted to the various representations of soil physical processes (e.g., the inclusion of vapor flow and air flow or not). See Line 166-172.

4. Results l172 and following : The numerical results in terms of computed evapotranspiration depend critically on the way to parameterize evapotranspiration, which is not presented in the paper. Various descriptions could be used here. For instance, and among many others, an empirical one emphasizing the role of vegetation could be find in Orgogozo et al., 2019, or a theoretically derived one in the case of purely evaporative processes could be find in Duval et al., 2004. The mathematical expressions and input data used to compute the evapotranspiration in each model should be described in the manuscript. Without these key information, it is not possible for the reader to interpret the given results.

**Response**: We used the Penman Monteith method to calculate the evapotranspiration and added the relevant description in the Sect. 2.2. The different soil physical processes alter the soil thermo-hydrological regimes then affect the actual surface evapotranspiration (see Line 156-166). The mathematical expressions are presented in Appendix A1.

5. Discussion l244 : The first sentence is wrong: the vapor transfer processes are not the only sources of couplings between thermal and hydrological transfers in porous media when freeze/thaw of the pore water occurs, see also my first comment on section 2. Methodology.

**Response**: Sorry for the confusion. Here we want to stress the important role of vapor transfer processes. Now rephrased as "Vapor flow, which is dependent on soil matric potential and temperature, links soil water and heat transfer processes." See Line 279.

6. ll254-l264 : Here is the explanation for the difference of amplitude of diurnal cycle between models. It seems to me that this is the key point of the discussion (evocated already numerous times in the manuscript, e.g;: l141, l147, l163), but somewhat hard to follow. It should be rewritten in a clearer way, may be with explicative schemes ?

**Response**: Many thanks for pointing out this. We made modifications by specifying the relevant figures and explicative schemes. See Line 294-310.

7. l274 : 'hydraulic conductivity' increase due to 'airflow from the atmosphere to the soil' ? Please give a short explanation.

**Response**: After precipitation events, the atmospheric humidity is high. Zeng et al. (2011a) verified that airflow (the air convection between atmosphere and topsoil) can bring the atmospheric moisture into topsoil. Thus the hydraulic conductivity of topsoil layers considerably increased. We rephrase as "since the hydraulic conductivity of topsoil layers increased tremendously due to the increased topsoil moisture by the injected airflow from the moist atmosphere". See Line 318.

8. l286-290 : The strong point made about evapotranspiration highlights the need to give to the reader all the relevant information about the handling of the evapotranspiration sink terms in each model (see also my comment for the section 3. Results). Please discuss also the transpirative component of evapotranspiration.

**Response**: We presented the descriptions of evapotranspiration, including the mathematical expressions (Appendix A1) and the relevant text (Line 149). Maqu experimental site is characteristic as seasonal frozen ground where growing the grassland. When soil freezes, grassland steps into the dormancy period. The dormancy period is ended when the integrated root zone soil temperature becomes positive. For our simulation period, soil started thawing only for a few days. The integrated soil temperature was not enough to break the vegetation dormancy. The transpiration thus has a very minimum effect during our simulation period. see Line 162-165.

9. l296-301 : The domain of applicability of the presented study should be better discussed. For instance, a point is made about the freeze/thaw mechanisms of permafrost while the studied field site is not in a permafrost affected area. The relative importance of the vapor flow, the thermal effect on water flow and the airflow should be more discussed with respect to the biogeoclimatic context (e.g. : more important in climate with long freezing/thawing periods or with long periods with surface temperature oscillating around 0_C), and in the context of the existing literature (e.g.: Karra et al., 2014).

**Response**: This part is for the outlook and applications of our work, which is to understand the freezing/thawing processes. The developed physical process-based model in this study can be applied to other frozen soil conditions. In the discussion, we made some extensions from frozen soils to permafrost. As well known, the active layer of permafrost region undergoes the freezing/thawing processes, which implicates the applicability of our model over the permafrost region. It is to note that the relative importance of the vapor flow, the thermal effect on water flow and airflow might vary among different regions under climate changing context. We discussed it a bit in Sect. 4.1 (Line 287-310).

10. Technical comments : The English language should be improved, although I am not a native English-speaking person so maybe I am making a mistake on that point. For instance it seems to me that the vegetation development cannot be 'closed' (l29). As another example I think that 'the best water resources management' or 'a better water resources management' could be used but not 'the better water resources management' (l33). A reread by an English editing service might be helpful.

**Response**: We made the corresponding changes (Line 28, 32) and have the manuscript English edited.

11. l40 and also in other places (e.g.: l42, l51) : Citations should be re-ordered (2006 before 2010).
**Response**: We re-ordered the citations by time and checked throughout the whole manuscript.

12. l138 : Fig1. The figure is not clear enough. Firstly it is difficult to decipher the different curves for 5, 10 and 20 cm depth – CPLD curve is nowhere visible (if it si beneath the CPLD-Air curve, make this

one discontinuous). The legend should also be clearer for the obs. Secondly I didn't got the 'earlier stepping in / stepping out of the frozen period', may be they should be pointed out in the figure itself.

**Response**: Thanks a lot. Figure 2 (original Figure 1) was replotted, with the dotted line for ACM-AIR curve, to make different curves visible. The solid line was used for the observation curve to make the legend clearer. We added the lines to indicate the "Freeze", "Transition", and "Thaw" periods in the Figure 2 to indicate the start/end dates of the frozen period.

13. l147 : Fig2. Same formal remarks that for Fig1.

**Response**: Figure 3 (original Figure 2) was replotted to highlight the differences.

Reference

Duval, F., Fichot, F., and Quintard, M.: A local thermal non-equilibrium model for two-phase flows with phase-change in porous media, International Journal of Heat and Mass Transfer, 47, 613-639, https://doi.org/10.1016/j.ijheatmasstransfer.2003.07.005, 2004.

Grenier, C., Anbergen, H., Bense, V., Chanzy, Q., Coon, E., Collier, N., Costard, F., Ferry, M., Frampton, A., Frederick, J., Gonçalvès, J., Holmén, J., Jost, A., Kokh, S., Kurylyk, B., McKenzie, J., Molson, J., Mouche, E., Orgogozo, L., Pannetier, R., Rivière, A., Roux, N., Rühaak, W., Scheidegger, J., Selroos, J. O., Therrien, R., Vidstrand, P., and Voss, C.: Groundwater flow and heat transport for systems undergoing freeze-thaw: Intercomparison of numerical simulators for 2D test cases, Adv Water Resour, 114, 196-218, 10.1016/j.advwatres.2018.02.001, 2018.

Karra, S., Painter, S. L., and Lichtner, P. C.: Three-phase numerical model for subsurface hydrology in permafrost-affected regions (PFLOTRAN-ICE v1.0), Cryosphere, 8, 1935-1950, 10.5194/tc-8-1935-2014, 2014.

Orgogozo, L., Prokushkin, A. S., Pokrovsky, O. S., Grenier, C., Quintard, M., Viers, J., and Audry, S.: Water and energy transfer modeling in a permafrost-dominated, forested catchment of Central Siberia: The key role of rooting depth, Permafr Periglac Proc, 30, 75-89, 10.1002/ppp.1995, 2019.

Zeng, Y., Su, Z., Wan, L., and Wen, J.: Numerical analysis of air-water-heat flow in unsaturated soil: Is it necessary to consider airflow in land surface models?, Journal of Geophysical Research: Atmospheres, 116, D20107, 10.1029/2011JD015835, 2011a.

**Referee #2:**

Frozen soil undergoing freeze-thaw cycles has significant impacts on local hydrology, ecosystems, and engineering infrastructure within the context of global warming. However, it is challenging to depict a dynamic thermal equilibrium system of ice, liquid water, water vapor and dry air in soil pores, when soil experiences the freeze/thaw process. Through careful design and analyses of numerical simulation experiments, this study may help us understand the contribution of airflow-induced water and heat transport in the frozen ground. I just have a few comments/suggestions that may improve this manuscript, before it can be accepted for publication in HESS.

We really appreciate your helpful comments on improving this manuscript.

Major comments:
1. The authors should clearly state/add the innovative points by this study in the title, abstract, and body text (e.g., objectives, results and discussions, as well as conclusions), by comparing to the listed publications in the references by the same authors. It is obvious that this group has quite a few nice publications on the physics of frozen ground, by describing the contributions/roles of vapor, liquid water and solid ice in the water and heat transports. After my reading of this manuscript, it is more like a sensitive study or a review paper. Please add text to clarify the major difference of this manuscript from previous studies, and demonstrate the new processes/knowledge to the permafrost hydrology community.

**Response**: The main purpose/motivation of this work is to understand the impact of various representations of soil physical processes on simulating hydrothermal regimes of frozen soils. Usually, such kind of investigation/inter-comparison is implemented via using different models, with different model frameworks, numerical solutions etc. In our study, we used an unified modeling framework, STEMMUS, to investigate the hydrothermal dynamics of frozen soil, considering uncoupled soil moisture and heat transfer (as in most of Land Surface Model), coupled soil moisture and heat transfer (via vapor flow), and further coupled with air transfer (i.e., both vapor and air flow). Such investigation with increasing levels of complexities in representing mass, momentum and energy transfer in frozen soil is the innovation of this study.

With the above approach, we can delineate the contributions of each individual process to the soil hydrothermal states, which can be further applied to figure out their roles in ecosystem response in cold regions. Furthermore, this study can also provide supports to define how complex the physical processes we should take into account when interpreting the hydrothermal regimes in cold regions. We added the relevant descriptions in introduction, discussion and conclusion part to highlight this.

2. In Figure (1-3 & 5), the red and blue lines are always overlapped. Is there a better way to show them?

**Response**: We replotted Figure 2-4 & 6 (original Figure 1-3 & 5), making the blue lines discontinuous, to make the difference visible.

3. The difference between CLPD-air and CLP is that air flow was taken into account. What are the key processes that the air flow affects frozen ground? The difference should be briefly introduced in Section 2.2, for better understanding in this manuscript.

**Response**: When considering air flow, air flow induced liquid and vapor flow and its corresponding heat flow were activated. While air flow coexists with vapor flow. The presence of air flow considerably affects the vapor transfer processes. The water and heat transfer in frozen soil are thus affected. All these aspects were briefly explained in Sect. 2.2 (Line 134-137).

4. There are many results in this paper, and I think you can add more details in Section 5 (conclusions).

**Response**: This paper mainly investigated the role of various soil physical processes in representing soil hydrothermal states and explain the underlying mechanisms. We find that the basic coupled model can not well capture the dynamics of soil moisture/temperature. Models with advanced coupled water and heat transfer processes largely improved its capability. The underlying reasons were analyzed via looking into the dynamics of heat budgets and subsurface latent heat flux density. We stressed the important role of vapor flow in the total mass and energy heat transfer during frozen periods and also the thermal effect on liquid flow. These physics contribute to a better soil temperature/moisture simulations by ACM (originally as CPLD).
Furthermore, the role of air flow was found only important along with vapor flow. The contribution of airflow to the total water and heat transfer is negligible. However, the consideration of air flow considerably affects the latent heat flux density and the heat transfer process especially during the freezing-thawing transition period.

We further added the description of other non-conductive heat fluxes (liquid/vapor/air induced convective heat fluxes) in conclusions. See Line 339.

5. Literature review about the frozen ground/permafrost hydrology by this manuscript is incomplete. I would like to suggest the authors also referring to the following ones but not limited to them. E.g.,

- *Qi et al. (2019). Coupled Snow and Frozen Ground Physics Improves Cold Region Hydrological Simulations: An Evaluation at the Upper Yangtze River Basin (Tibetan Plateau). Journal of Geophysical Research: Atmospheres, 124(33): 12985-13004.*

- *Biskaborn et al. (2019).Permafrost is warming at a global scale. Nature communications, 10(1), 264.*

- *Wang et al. (2017). Development of a land surface model with coupled snow and frozen soil physics. Water Resources Research, 53, 5085–5103.*

- *Bao et al. (2016). Development of an enthalpy-based frozen soil model and its validation in a cold region in China. Journal of Geophysical Research: Atmospheres, 121(10), 5259-5280.*

- *Iijima, Y., Ohta, T., Kotani, A., Fedorov, A.N., Kodama, Y., & Maximov, T.C. (2014). Sap flow changes in relation to permafrost degradation under increasing precipitation in an eastern Siberian larch forest. Ecohydrology, 7(2), 177-187.*

**Response**: We added more in the literature review about the frozen ground/permafrost hydrology. Meanwhile stress the novelty/importance of our study. See Sect. 1 Introduction (Line 46-50; 73-79).

**Understanding the Mass, Momentum and Energy Transfer in the Frozen Soil with Three Levels of Model Complexities**

Lianyu Yu[1], Yijian Zeng[1], Zhongbo Su[1,2]

[1] Faculty of Geo-information and Earth Observation (ITC), University of Twente, Enschede, The Netherlands

[2] Key Laboratory of Subsurface Hydrology and Ecological Effect in Arid Region of Ministry of Education, School of Water and Environment, Chang'an University, Xi'an, China

*Correspondence to*: Yijian Zeng (y.zeng@utwente.nl); Zhongbo Su (z.su@utwente.nl)

**Abstract**

Frozen ground covers a vast area of earth surface and has its important ecohydrological implications for cold regions under changing climate. However, it is challenging to characterize the simultaneous transfer of mass and energy in frozen soils. Within the modeling framework of STEMMUS (Simultaneous Transfer of Mass, Momentum and Energy in Unsaturated Soil), the model complexity of soil heat and mass transfer varies from the basic coupled (termed as BCM), to the advance coupled heat and mass transfer (ACM), and further to the explicit consideration of airflow (ACM-AIR).  The impact of different model complexities on understanding the mass, momentum and energy transfer in frozen soil was investigated. The model performance in simulating water and heat transfer and surface latent heat flux was evaluated over  a typical Tibetan Plateau meadow site. Results indicate that the ACM considerably improved the simulation of soil moisture, temperature and latent heat flux. The analyses of heat budget reveal that the improvement of soil temperature simulations by ACM is attributed to its physical consideration of vapor flow and thermal effect on water flow, with the former mainly functions above the evaporative front and the latter dominates below the evaporative front. The contribution of airflow-induced water and heat transport (driven by the air pressure gradient) to the total mass and energy fluxes is negligible. Nevertheless, given the explicit consideration of airflow, vapor flow  and its effect on heat transfer were enhanced during the freezing-thawing transition period.

**1. Introduction**

Frozen soils have been reported with significant changes under climate warming (Cheng and Wu, 2007;Hinzman et al., 2013;Biskaborn et al., 2019;Zhao et al., 2019). Changes in the freezing/thawing process can alter soil hydrothermal regimes and the water flow pathways and thus affect vegetation development (Walvoord and Kurylyk, 2016). Such changes will further considerably affect the spatial pattern, the seasonal to interannual variability and long term trends in land surface water, energy and carbon budgets and then the land  surface-atmosphere interactions (Subin et al., 2013;Iijima et al., 2014;Schuur et al., 2015;Walvoord and Kurylyk, 2016). Understanding the soil freeze/thaw processes appears to be the necessary path  for a better water resources management and ecosystem protection in cold regions.

When soil experiences the freeze/thaw process, there is a dynamic thermal equilibrium system of ice, liquid water, water vapor and dry air in soil pores. Water and heat flow are tightly coupled in frozen soils. Coupled water and heat physics, describing the concurrent flow of liquid, vapor as well as heat flow, was first proposed by Philip and De Vries (1957) (hereafter termed as PdV57), considering the enhanced vapor transport. The PdV57 theory has been widely applied for a detailed understanding of soil evaporation during the drying process (De Vries, 1958;Milly, 1982;De Vries, 1987;Saito et al., 2006;Novak, 2010). The attempts to simulate the coupled water and heat transport in frozen soils started in 1970s (e.g., Harlan, 1973;Guymon and Luthin, 1974). Since then, numerical tools for simulating one-dimensional frozen soil were  gradually developed. Flerchinger and Saxton (1989) developed the SHAW model with the capacity of simulating the coupled water and heat transport process. Hansson et al. (2004) accounted for the phase changes in HYDRUS-1D model and verified its numerical stability with rapidly changing boundary conditions. Considering the two components (water and gas) and three water phases (liquid, vapor, and solid), Painter (2011) developed a fully coupled water and heat transport model MarsFlo. Aiming to efficiently deal with the water phase change between liquid and ice, the enthalpy-based frozen soil model (using enthalpy and total water mass instead of temperature and liquid water content as the prognostic variables) was developed and demonstrated its capability to stably and efficiently simulate soil freeze/thaw process (Li et al., 2010;Bao et al., 2016;Wang et al., 2017). These works together with other modifications, simplifications, generate  a hierarchy of frozen soil models, see the detail review by Li et al. (2010) and Kurylyk and Watanabe (2013).

Air flow has been reported important to the soil water and heat transfer process under certain conditions (Touma and Vauclin, 1986;Prunty and Bell, 2007). Zeng et al. (2011a, b) found that soil evaporation is enhanced after precipitation events by considering air flow and demonstrated that the air pressure induced advective fluxes inject the moisture into the surface soil layers and increase the hydraulic conductivity at the top layer. The diurnal variations of air pressure resulted in the vapor circulation between the atmosphere and

the land surface. Wicky and Hauck (2017) reported that the temperature difference between the upper and the lower part of a permafrost talus slope was significant and attributed it to the airflow induced convective heat flux. Yu et al. (2018) analyzed the spatial and temporal dynamics of air pressure induced fluxes and

65    found an interactive effect as the presence of soil ice. The abovementioned studies demonstrate that the explicit consideration of air flow has the potential to affect the soil hydrothermal regime. However, to what extent and under what condition air flow plays significant roles in the subsurface heat budgets has not been detailed.

Current land surface models (hereafter LSMs), however, usually adopted a simplified frozen soil physics

70    with relative coarse vertical discretization (Koren et al., 1999;Viterbo et al., 1999;Niu et al., 2011;Swenson et al., 2012). In their parameterizations, soil water and heat interactions can only be indirectly activated by the phase change processes, the mutual dependence of liquid water, water vapor, ice and dry air in soil pores is  absent. This mostly lead to oversimplifications of physical representations of hydrothermal and ecohydrological dynamics in

75    cold regions (Novak, 2010;Su et al., 2013;Wang et al., 2017;Cuntz and Haverd, 2018;Grenier et al., 2018;Wang and Yang, 2018;Qi et al., 2019). Specifically, Su et al. (2013) evaluated the European Centre for Medium-Range Weather Forecasts (ECMWF) soil moisture analyses  over the Tibetan Plateau, and found that HTESSEL cannot capture phase transitions of soil moisture (i.e., underestimation during frozen

80    period while overestimation during thawing). There are continuous efforts in improving parameterizations and representations of cold region dynamics, including frozen ground (Boone et al., 2000;Luo et al., 2003), vapor diffusion (Karra et al., 2014), thermal diffusion (Bao et al., 2016), coupling water and heat transfer (Wang and Yang, 2018), and three-layer snow physics (Wang et al., 2017;Qi et al., 2019). While to our knowledge, few studies have investigated the role of increasing complexities of soil physical processes (from

85    the basic coupled to the advanced coupled water and heat transfer processes, and then the explicit consideration of air flow) in simulating the thermo-hydrological states in cold regions. How and to what extent the complex mutual dependence physics affects the soil mass and energy transfer in frozen soils? Is it necessary to consider  a fully coupled physical process in LSMs? These two questions frame the scope of this work.

90    In this paper, we incorporated  various complexities of soil water and heat transport mechanisms into a common modeling framework (STEMMUS-FT, Simultaneous Transfer of Energy, Momentum and Mass in Unsaturated Soils with Freeze-Thaw). With the aid of in situ measurements collected from a typical Tibetan meadow site, the pros and cons of different model complexities were investigated. Subsurface energy budgets and latent heat flux density analyses were further carried out to

95    illustrate the underlying  mechanisms of different coupled soil water-heat physics. Section 2 describes the experimental site and three different complexities

complexity of subsurface physics into within the STEMMUS framework. Performance The performance of different models is presented in Section 3 together with the subsurface heat budgets and latent heat flux density analyses. Section 4 discusses the effects of considering coupled soil water-heat transport transfer and air flow in frozen soils. The Cconclusion is made drawn in Section 5.

**2. Methodology**

**2.1 Experimental site**

Maqu station, equipped with a catchment catchment-scale soil moisture and soil temperature (SMST) monitoring network and micro-meteorological observing system (Su et al., 2011;Dente et al., 2012;Zeng et al., 2016), is situated on the north-eastern fringe of the Tibetan Plateau (33°30'–34°15'N, 101°38'–102°45'E). According to the updated Köppen-Geiger climate classification system, it can be characterized as a cold climate with dry winter and warm summer (Dwb). The mean annual air temperature is 1.2 ℃, and the mean air temperatures of the coldest month (January) and warmest month (July) are about -10.0 ℃ and 11.7 ℃, respectively. Alpine meadows (e.g., *Cyperaceae* and *Gramineae*), with heights varying from 5 cm to 15 cm throughout the growing season, are the dominant land cover in this region. In situ soil sampling determined the soil as a mixture ofThe sandy loam and, silt loam are found by in situ soil sampling and organic soil with a maximum of 18.3 % organic matter for the upper soil layers (Dente et al., 2012;Zheng et al., 2015a;Zhao et al., 2018).

The Maqu SMST monitoring network spans an area of approximately 40 km×80 km with the elevation ranging from 3200 m to 4200 m a.s.l. SMST profiles are automatically measured by 5TM ECH$_2$O probes (METER Group, Inc., USA) installed at the soil depths of 5 cm, 10 cm, 20 cm, 40 cm, and 80 cm. The micro-meteorological observing system includes a 20 m Planetary Boundary Layer (PBL) tower providing the meteorological measurements at five heights above ground (i.e., wind speed and direction, air temperature and relative humidity), and an eddy-covariance (EC150, Campbell Scientific, Inc., USA) system installed for measuring the turbulent sensible, latent heat fluxes and carbon fluxes. Four component down and upwelling solar and thermal radiation (NR01-L, Campbell Scientific, Inc., USA), and liquid precipitation (T200B, Geonor, Inc., USA) are also monitored.

**2.2 Mass and energy transport in unsaturated soils**

On the basis of STEMMUS modelling framework, the increasing complexity of vadose zone physics in frozen soils was implemented as three alternative models (Table 1). Firstly, STEMMUS enabled the isothermal water and heat transfer physics (Eqs. 1 & 2). The 1-D Richards equation is utilized to solve the isothermal water transport in variably saturated soils. The heat conservation equation took into account the freezing/thawing process and the latent heat due to water phase change. The effect of soil ice on soil hydraulic

and thermal properties was considered. It is termed as basic coupled water and heat transfer model (BCM).

Secondly, the fully coupled water and heat physics, i.e., water vapor flow and thermal effect on water flow, was explicitly considered in STEMMUS, termed as the advanced coupled model (ACM). For the  ACM physics, the extended version of Richards (1931) equation with modifications made by Milly (1982) was used as the water conservation equation (Eq. 3). Water flow can be expressed as liquid and vapor fluxes driven by both temperature gradient and matric potential gradient. The heat transport in frozen soils mainly includes: heat conduction (CHF, $\lambda_{eff}\frac{\partial T}{\partial z}$ ), convective heat transferred by liquid flux (HFL, $-C_L q_L(T - T_r)$, $-C_L S(T - T_r)$ ), vapor flux (HFV, $-C_V q_V(T - T_r)$ ), the latent heat of vaporization (LHF, $-q_V L_0$  ), the latent heat of freezing/thawing ($-\rho_i \theta_i L_f$) and a source term associated with the exothermic process of wetting of a porous medium (integral heat of wetting) ($-\rho_L W \frac{\partial \theta_L}{\partial t}$). It can be expressed as Eq. 4 (De Vries, 1958;Hansson et al., 2004).

Lastly, STEMMUS expressed the freezing soil porous media as the mutual  dependent system of liquid water, water vapor, ice water, dry air and soil grains, in which other than air flow all other components kept the same as in ACM, termed as ACM-AIR model (Eqs. 5, 6, &7, Zeng et al., 2011a, b;Zeng and Su, 2013). The effects of air flow on soil water and heat transfer can be two-fold. Firstly,  the air flow induced water and vapor fluxes ($q_{La}$, $q_{Va}$) and its corresponding convective heat flow (HFa, $q_a C_a(T - T_r)$) were considered. Secondly, the presence of air flow alters the vapor transfer processes, thus can considerably affects the water and heat transfer in an indirect manner.

 STEMMUS utilized the adaptive time step strategy, with maximum time steps ranging from 1s to 1800s (e.g., with 1800s as the time step under stable conditions). The maximum desirable change of soil moisture and soil temperature within one time step was set as 0.02 cm³ cm⁻³ and 2 °C, respectively, to prevent too large change in state variables that may cause numerical instabilities. If the changes between two adjacent soil moisture/temperature states are less than the maximum desirable change, STEMMUS continues without changing the length of current time step (e.g., 1800s). Otherwise, STEMMUS will adjust the time step with a deduction factor, which is proportional to the difference between the too large changes and desirable allowed maximum changes of state variables. Within one single time step, the Picard iteration was used to solve the numerical problem, and the numerical convergence criteria is set as 0.001 for both soil matric potential (in cm) and soil temperature (in °C).

To accommodate the specific conditions of a Tibetan meadow, the total depth of the soil column was set as

1.6 m (Figure 1). The vertical soil discretization was designed finer for the upper soil layers (0.1-2.5 cm for 0-40 cm, 27 layers) than that for the lower soil layers (5-20 cm for 40-160 cm, 10 layers).  Surface boundary for the water transport was set as the flux-type boundary controlled by the atmospheric forcing (i.e., evaporation, precipitation) while the specific soil temperature was assigned as the surface boundary of the energy conservation equation. The free drainage (zero matric potential gradient) and measured soil temperature were set as the bottom boundary conditions for the water transport and heat transport, respectively. For the air flow, the surface boundary was set as the atmospheric pressure and soil air was allowed to escape from the bottom of the soil column. Surface evapotranspiration was calculated using the Penman-Monteith method. Soil evaporation and transpiration can be separately estimated. The available radiation energy is partitioned into the canopy and soil component via LAI, the canopy minimum surface resistance and soil surface resistance are then utilized to calculate the potential transpiration and soil evaporation. Actual transpiration is calculated as the function of potential transpiration and the root length density-weighted available soil liquid water (which is assumed to be zero if soil temperature falls below 0 °C (Kroes et al., 2009;Orgogozo et al., 2019)). For our simulation period, grassland stepped into the dormancy period as the soil freezes. The accumulative positive temperature during the thawing period was not enough to break the dormancy of vegetation. The contribution of plant transpiration to the land surface heat flux is negligible in the dormancy period. The effect of soil moisture on the actual soil evaporation is taken into account via the soil surface resistance (Eq. A6). All three aforementioned models adopted the same adaptive time-step strategy and numerical solution, the same soil discretization, soil parameters (shown as Table 2) and boundary conditions. Since all three models employed the same mesh resolutions, parameters and boundary conditions, numerical solution and utilized the adaptive time step strategy. It indicated that the truncation errors due to numerical solution among three models were comparable. The difference among models is mainly restricted to the various representations of soil physical processes (e.g., the inclusion of vapor flow and air flow or not).

**3. Results**

Given  the same atmospheric forcing and the same set of parameters, the performance of models with varying  complexities of  soil water and heat physics was illustrated  in Sect. 3.1, 3.2 & 3.3. Sect. 3.4 & 3.5 further analyzed the variations of heat budgets and subsurface latent heat flux density, illustrating differences the underlying  mechanisms among various models.

**3.1 Soil hydrothermal profile simulations**

195    The performance of the model with various soil physics in simulating the soil thermal profile information is illustrated in Figure 2. Both ACM and ACM-AIR  well reproduced the time series of the soil temperature at different soil depth except for the 40 cm, which is most probably due to the inappropriate measurements (e.g., improper placement of sensors). However, there are significant discrepancies in soil temperature simulated by the BCM. Compared to the observations,

200    a stronger diurnal behavior of soil temperature in response to the fluctuating atmospheric forcing was found and  earlier stepping-in/stepping-out of the frozen period was simulated by the BCM. Such differences enlarged at deeper soil layers with large BIAS and RMSE values (Table 3).

Figure 3 presents the time series of observed and simulated soil liquid water content at five soil

205    layers. During the rapid freezing period, a noticeable overestimation of diurnal fluctuations and early and fast decreasing of soil liquid water content was simulated by BCM. Moreover,  strong diurnal fluctuations and early increase of liquid water content were also found during the thawing period. The early thawing of soil water even led to an unrealistic refreezing process at 80 cm (from 88[th] to 92[nd] day after December 2015), which is due to the simulated early warming of soil by BCM (Figure

210    2). Such discrepancies were significantly ameliorated  by ACM and ACM-AIR simulations. Nevertheless, all three models can well capture the diurnal variations and magnitude of liquid water content during the frozen period. Note that there is an observable difference between ACM and ACM-AIR simulated soil liquid water content at shallower soil layers during the thawing process (e.g., Figure 3, 5cm).

215    ### 3.2 Freezing front propagation

The time series of freezing front propagation derived from the measured and simulated soil temperature was reproduced  in Figure 4. Initialized from the soil surface, the freezing front quickly develops downwards till the maximum freezing depth. The thawing process starts from both the top and bottom, mainly driven by the atmospheric heat and bottom soil temperature,

220    respectively. Such characteristics were well captured by both the ACM and ACM-AIR model in terms of freezing rate, maximum freezing depth and surface thawing process while the BCM tended to present a more fluctuated and rapid freezing front propagation and a deeper maximum freezing depth  that is early reached. The effect of atmospheric heat source on soil  was overestimated by the BCM as shown by the stronger diurnal early onset of the thawing

225    process.

**3.3 Surface Evapotranspiration**

The performance of the model with different soil physics in reproducing the latent heat flux dynamics is shown in Figure 5. Compared to the observed LE, there is a significant overestimation of half-hourly latent heat flux, which significantly degraded the overall performance using BCM. The occurrence of such overestimation was notably reduced using ACM and ACM-AIR. The general underestimation of latent heat flux by the ACM and ACM-AIR was found mostly during the freezing-thawing transition period (Figure 6b), when the soil hydrothermal states are not well captured (Figure 2 & 3).

The overestimation of surface evapotranspiration by BCM was significant during the initial freezing and transition period (Figure 6a, December & February). During the rapid freezing period (January), BCM presented a good match in the diurnal variation compared to the observations. The monthly average diurnal variations were found to be well captured by ACM and ACM-AIR. Figure 6b shows the comparison of observed and simulated cumulative surface evapotranspiration. The overall overestimation of surface evapotranspiration by BCM can be clearly seen in Figure 6b. Days at the initial freezing periods, with high liquid water content simulations, accounted for more than 90% of the overestimation. The initial stage overestimation of surface evapotranspiration was significantly reduced by ACM and ACM-AIR. Slight underestimation of cumulative surface evapotranspiration was simulated by ACM and ACM-AIR with values of 3.98% and 4.78%, respectively.

**3.4 Heat budgets**

Figure 7 shows the time series of simulated energy budget components at 5 cm using BCM, ACM and ACM-AIR during the freezing period (5[th] - 11[th] day after 1 December) and freezing-thawing transition period (83[rd] - 89[th] day after 1 December). For the BCM, only the change rate of heat content HC and conductive heat flux divergence CHF are considered as the LHS and RHS of Eq. 2 (see Table 1). Three additional terms, convective heat flux divergence of liquid flow HFL and vapor flow HFV, and latent heat flux divergence were included for the ACM. While for the ACM-AIR, the convective heat flux divergence of air flow HFa was further added.

There is a strong diurnal variation of heat budget components (HC, CHF & LHF, Table 1), corresponding to the diurnal fluctuation of soil temperature. For the BCM, the change rate of heat content was almost completely balanced by the conductive heat flux divergence CHF (Figure 7a). Compared to the BCM, a stronger diurnal fluctuation of HC and CHF was found in ACM results. Inferred from results in Figure 2, the time series of soil temperature change regarding to time ($\partial T/\partial t$) simulated by BCM was larger

260  than that simulated by ACM. This indicates BCM produced  less

fluctuation of apparent heat capacity  ($C_{app} = C_{soil} + \rho_i \frac{L_f^2}{gT}\frac{d\theta_L}{d\psi}$) than ACM. During the

freezing period, the latent heat flux divergence LHF was lower than conductive heat flux divergence CHF by

1-2 orders of magnitude (Figure 7b). The positive value of LHF term during daytime indicates condensation

happens at 5 cm, as water vapor moves downward . The convective heat fluxes of liquid

265  flow and vapor flow  were even smaller compared to conductive heat flux (Figure 7b). There is no

significant difference of heat budget components between ACM and ACM-AIR  in terms

of diurnal variation and magnitude. The convective heat flux divergence of air flow played a negligible role

in the change of thermal state (HC) (Figure 7c).

The dynamics of heat balance components at 5 cm soil layer was  simulated for the freezing-thawing

270  transition period (Fig.ure 7-d, e, f). Both HC and CHF underwent strong diurnal variations with increasing

fluctuation magnitude, indicating soil  warming at 5 cm . For the

ACM, CHF  outnumbered HC during daytime and the difference increased with time. Negative

values were found for LHF and developed further over time. The sum of CHF and LHF

nearly balance the HC term. Such behavior was similarly reproduced by ACM-AIR  with a

275  slightly large difference between HC and CHF terms. This means a larger amount of water vapor was

evaporated from 5 cm soil layer (with more negative LHF term) from ACM-AIR simulations than that

from ACM simulations, which explains the lower liquid water content for ACM-AIR

(Fig.ure 3, 5 cm).

**3.5 Subsurface latent heat flux density**

280  To give more context to the results, the spatial and temporal distribution of  simulated latent heat flux

density ($S_h$), $-\rho_w L \partial q_v / \partial z$, during the freezing and freezing-thawing transition period  were shown  in

Fig.ure 8. For the BCM, the latent heat flux density ($S_h$) is not available

as it neglects the vapor flow .

Figure  8a shows that there is a strong diurnal variation of $S_h$ at upper 0.1 cm soil layers. Such diurnal

285  behavior along the soil profile was interrupted at 1 cm, at which the water vapor consistently

moved upwards as evaporation source (termed as evaporative front). The path of this upward water vapor

 was disrupted at  20 cm from the 6th of December, where the freezing front developed.

Compared to the upper 0.1 cm soil, a weaker diurnal fluctuation of $S_h$ was found at lower soil layers.  For

ACM-AIR , the vapor transfer patterns  were similar to that of ACM (Fig.

290  8b). There were isolated connections of condensed water vapor between upper 1 cm soil and the lower soil

layers ($S_h$>0, e.g., 6th, 7th, 9th, and 10th of December), possibly associated with the downward air flow (see

Fig. 12 in Yu et al. (2018)). The large difference in magnitude of latent heat flux density between

ACM and ACM-AIR  appeared mainly isolated at upper soil layers (Fig.8c). At soil layers between 1 cm and 20 cm, ACM-AIR  simulated less in condensation vapor area ($S_h>0$) and more in the evaporation area ($S_h<0$), indicating that ACM-AIR  produced an additional amount of condensation and evaporation water vapor compared with ACM (Fig.8c).

Similar to that during the freezing period, the $S_h$ during the transition period can be characterized as:  strong diurnal variations at upper soil layers; interruption of diurnal patterns by the constant upward evaporation of intermediate soil layers; and weak diurnal variation at lower soil layers  (Fig.7d, e). While the maximum evaporation rate was less than that during the freezing period. The consistent evaporation zone developed to a depth of 5 cm. The path for the upwards water vapor tended to develop deeper than 30 cm with the absence of soil ice. The simulation by ACM-AIR  produced more condensation and less  evaporative water vapor than that by ACM  (Fig.8f). In addition, steadily more  evaporative water vapor from  5 cm was simulated by ACM-AIR  compared to ACM. This confirms the aforementioned point that during the freezing-thawing transition period, large LHF values were simulated by ACM-AIR  (Fig 67).

**4. Discussion**

**4.1 Coupled Water and Heat Transfer Processes**

Vapor flow, which is dependent on soil matric potential and temperature, links soil water and heat transfer processes.  The mutual dependence of soil water, in different phases (liquid, water vapor, and ice), and heat transport is enabled to facilitate our better understanding of the complex soil physical processes (e.g., Fig 67-78). Specifically, the interdependence of soil moisture and soil temperature (SMST) profiles simulated by  ACM was closer to the observation than that by CPLD model. In addition, significant enhancement in portraying the monthly average diurnal variations of surface evapotranspiration and cumulative evapotranspiration can be found from ACM simulations, which constrains the hydrothermal regimes especially during the freezing-thawing transition periods (Fig.1, 23& 65).

During the freezing period, liquid water in the soil freezes, which is analog to the soil drying process, and water vapor fluxes instead of liquid fluxes dominate the mass transfer process (Zhang et al., 2016). Neglecting such important water flux component unavoidably results in different/unrealistic simulations of surface evapotranspiration and SMST profiles (Li et al., 2010;Karra et al., 2014;Wang and Yang, 2018). Li et al. (2010) reported that vapor fluxes were comparable to the liquid water fluxes and affected the freezing/melting processes. On the basis of long term one-dimensional soil column simulations, Karra et al. (2014) reported that the inclusion of the vapor diffusion effect significantly increased the thickness of the ice layer as

explained by the positive vapor cold trapping-thermal conductivity feedback mechanism. From the energy budget perspective,  latent heat fluxes contribute more, due to the vapor phase change (LHF),  to the heat balance budget  at soil layers above the evaporative front than that below it (see LHF in Figure  7e vs. Figure 7b, corresponding evaporative front shown as Figure  8d vs. Figure 8a). This is consistent with findings by Zhang et al. (2016), who presented that the latent heat of vapor due to phase change is the two orders of magnitude less than the heat fluxes due to conduction during winter time and corresponds to our results of Figure 7b & c during the freezing period. While our results further showed that the latent heat fluxes due to vapor phase change can be considerable during the transition period (Figure 7e &f). The downward latent heat flux from ACM makes the subsurface soil warmer, which reduces the temperature gradient $(\partial T/\partial z)$ (Wang and Yang, 2018). This further results in the weaker diurnal fluctuations of  HC term  in  ACM  than that  in the BCM (see HC in Figure 7e vs. Figure 7d). At the soil layers below the evaporative front, the heat flux source from the vapor  transfer process (LHF) is negligible (e.g., Figure 7b). The thermal retard effect as the presence of soil ice, expressed as the apparent heat capacity term ($C_{app}$), dominates the heat transfer process in frozen soils. By considering the thermal effect on water flow, ACM usually has a larger water capacity value $\partial\theta/\partial\psi$ than BCM does. As such, the intense thermal impedance effect leads to the results that ACM  produced a weaker diurnal fluctuation of soil temperature than BCM at subsurface soil layers (e.g., Figure 2, 20 cm).

**4.2 Air Flow in the Soil**

Since soil pores are filled with liquid water, vapor and dry air, taking dry air as an independent state variable can facilitate a better understanding of the relative contribution of each component  to the mass and heat transfer in soils. The results show that the dry air-induced water and heat flow is negligible to the total mass and energy transfer (Zeng et al., 2011b;Yu et al., 2018). Nevertheless, dry air can affect soil hydrothermal regimes significantly under certain circumstances. Wicky and Hauck (2017) reported that the airflow-induced convective heat transfer resulted in a considerable temperature difference between the upper and lower part of a permafrost talus slope and thus  had a remarkable effect on the thermal regime of the talus slope. Zeng et al. (2011b) demonstrated the airflow-induced surface evaporation enhanced after precipitation events, since the hydraulic conductivity of topsoil layers increased tremendously due to the increased topsoil moisture by the injected airflow from the moist atmosphere . In this study, we found that the explicit consideration of airflow introduced an additional amount of subsurface condensation and evaporative water vapor in the condensation region and evaporation region, respectively (Figure 78c & f). The effect of latent heat flux on heat transfer was enhanced by airflow during the freezing-thawing transition period (Figure 67), which further  affected the subsurface hydrothermal simulations (e.g., Figure 23).

**5. Conclusions**

On the basis of STMMUS modeling framework with various  representations of water and heat transfer physics (BCM, ACM and ACM-AIR ), the performance of each model in simulating water and heat transfer and surface evapotranspiration was evaluated over a typical Tibetan meadow ecosystem. Results indicate that compared to  in situ observations, the BCM tended to present  earlier freezing and thawing date with a stronger diurnal variation of soil temperature/liquid water in response to the atmospheric forcing. Such discrepancies were considerably reduced by the model with the advanced coupled water-heat physics. Surface evapotranspiration was overestimated by BCM, mainly due to the mismatches during the initial freezing and freezing-thawing transition period. ACM models, with the coupled constraints from the perspective of water and energy conservation, significantly improve the model performance in mimicking the surface evapotranspiration dynamics during the frozen period. The analysis of heat budget components and latent heat flux density revealed that the improvement of soil temperature simulations by ACM is ascribed to its physical consideration of vapor flow and thermal effect on water flow, with the former mainly functions at regions above the evaporative front, and the latter dominates below the evaporative front. The non-conductive heat processes (liquid/vapor/air induced heat convection flux) contributed very minimal to the total energy fluxes during the frozen period except the latent heat flux divergence at the topsoil layers. The contribution of airflow induced water and heat flow to the total mass and energy fluxes is negligible. However, given the explicit consideration of airflow, the latent heat flux and its effect on heat transfer were enhanced during the freezing-thawing transition period. This work highlighted the role of considering the vapor flow, thermal effect on water flow, and airflow in portraying the subsurface soil hydrothermal dynamics, especially during freezing-thawing transition periods. To sum up, this study can contribute to a better understanding of freeze-thaw mechanisms of frozen soils, which will subsequently contribute to the quantification of permafrost carbon feedback (Burke et al., 2013;Kevin et al., 2014;Schuur et al., 2015), if the STEMMUS-FT model is to be coupled with a biogeochemical model, as lately implemented (Yu et al., 2020).

*Data availability.* The soil hydraulic/thermal property data can be freely downloaded from 4TU. Center for Research Data (https://doi.org/10.4121/uuid:61db65b1-b2aa-4ada-b41e-61ef70e57e4a). Some relevant data are made available from 4TU. Center for Research Data (https://doi.org/10.4121/uuid:cc69b7f2-2448-4379-b638-09327012ce9b).

*Author contribution.* Conceptualization, Z.S. and Y.Z.; methodology, L.Y., Y.Z.; writing - original draft preparation, L.Y., Y.Z.; writing – review & editing, L.Y.,Y.Z., Z.S..

395    *Competing interests.* The authors declare that they have no conflict of interest.

**Acknowledgment**

This work is supported by the National Natural Science Foundation of China (grant no. 41971033) and supported by the Fundamental Research Funds for the Central Universities, CHD (grant no. 300102298307).
400    The authors thank the editor and referees very much for their constructive comments on improving the manuscript.

**Appendix**

**A1. Calculation of surface evapotranspiration**

405    The one step calculation of actual soil evaporation ($E_s$) and potential transpiration ($T_p$) is achieved by incorporating canopy minimum surface resistance and actual soil resistance into the Penman-Monteith model (i.e., the $ET_{dir}$ method in Yu et al. (2016)). LAI is implicitly used to partition available radiation energy into the radiation reaching the canopy and soil surface.

$$T_p = \frac{\Delta(R_n^c - G) + \rho_a c_p \frac{(e_s - e_a)}{r_a^c}}{\lambda\left(\Delta + \gamma\left(1 + \frac{r_{c,min}}{r_a^c}\right)\right)} \tag{A1}$$

$$E_s = \frac{\Delta(R_n^s - G) + \rho_a c_p \frac{(e_s - e_a)}{r_a^s}}{\lambda\left(\Delta + \gamma(1 + \frac{r_s}{r_a^s})\right)} \tag{A2}$$

where $R_n^c$ and $R_n^s$ (MJ m$^{-2}$ day$^{-1}$) are the net radiation at the canopy surface and soil surface, respectively; $\rho_a$
410    (kg m$^{-3}$) is the air density; $c_p$ (J kg$^{-1}$ K$^{-1}$) is the specific heat capacity of air; $r_a^c$ and $r_a^s$ (s m$^{-1}$) are the aerodynamic resistance for canopy surface and soil surface, respectively; $r_{c,min}$ (s m$^{-1}$) is the minimum canopy surface resistance; and $r_s$ (s m$^{-1}$) is the soil surface resistance.

The net radiation reaching the soil surface can be calculated using the Beer's law:

$$R_n^s = R_n \, exp(-\tau LAI) \tag{A3}$$

And the net radiation intercepted by the canopy surface is the residual part of total net radiation:

$$R_n^c = R_n(1 - exp(-\tau LAI)) \tag{A4}$$

415    The minimum canopy surface resistance $r_{c,min}$ is given by:

$$r_{c,min} = r_{l,min}/LAI_{eff} \tag{A5}$$

where $r_{l,min}$ is the minimum leaf stomatal resistance; $LAI_{eff}$ is the effective leaf area index, which considers that generally the upper and sunlit leaves in the canopy actively contribute to the heat and vapor transfer.

The soil surface resistance can be estimated following van de Griend and Owe (1994).

$$r_s = r_{sl} \qquad \qquad \theta_1 > \theta_{min}, h_1 > -100000 \ cm$$

$$r_s = r_{sl}e^{a(\theta_{min}-\theta_1)} \quad \theta_1 \leq \theta_{min}, h_1 > -100000 \ cm \qquad \qquad (A6)$$

$$r_s = \infty \qquad \qquad h_1 \leq -100000 \ cm$$

where $r_{sl}$ (10 s m$^{-1}$) is the resistance to molecular diffusion of the water surface; $a$ (0.3565) is the fitted parameter; $\theta_1$ is the topsoil water content; $\theta_{min}$ is the minimum water content above which soil is able to deliver vapor at a potential rate.

The root water uptake term described by Feddes et al. (1978) is:

$$S(h) = \alpha(h)S_p \qquad \qquad (A7)$$

where $\alpha(h)$ (dimensionless) is the reduction coefficient related to soil water potential $h$; and $S_p$ (s$^{-1}$) is the potential water uptake rate.

$$S_p = b(z)T_p \qquad \qquad (A8)$$

where $b(z)$ is the normalized water uptake distribution, which describes the vertical variation of the potential extraction term, $S_p$, over the root zone. Here the asymptotic function was used to characterize the root distribution as described in (Gale and Grigal, 1987;Jackson et al., 1996;Yang et al., 2009;Zheng et al., 2015b). $T_p$ is the potential transpiration in (A1).

430 **Notation**

| Symbol | Parameter | Unit | Value |
|---|---|---|---|
| $a$ | Fitted parameter for soil surface resistance | - | 0.3565 |
| $b(z)$ | Normalized water uptake distribution | m$^{-1}$ | |
| $C_a$ | Specific heat capacity of dry air | J kg$^{-1}$ °C$^{-1}$ | 1.005 |
| $C_{app}$ | Apparent heat capacity | J kg$^{-1}$ °C$^{-1}$ | $= C_{soil} + \rho_i \dfrac{L_f^2}{gT} \dfrac{d\theta_L}{d\psi}$ |
| $C_i$ | Specific heat capacity of ice | J kg$^{-1}$ °C$^{-1}$ | 2.0455 |
| $C_L$ | Specific heat capacity of liquid | J kg$^{-1}$ °C$^{-1}$ | 4.186 |
| $C_s$ | Specific heat capacity of soil solids | J kg$^{-1}$ °C$^{-1}$ | |
| $C_{soil}$ | Heat capacity of the bulk soil | J kg$^{-1}$ °C$^{-1}$ | |
| $C_V$ | Specific heat capacity of water vapor | J kg$^{-1}$ °C$^{-1}$ | 1.87 |
| $c_p$ | Specific heat capacity of air | J kg$^{-1}$ K$^{-1}$ | |
| $D_e$ | Molecular diffusivity of water vapor in soil | m$^2$ s$^{-1}$ | |
| $D_{TD}$ | Transport coefficient for adsorbed liquid flow due to temperature gradient | kg m$^{-1}$ s$^{-1}$ °C$^{-1}$ | |
| $D_{Va}$ | Advective vapor transfer coefficient | s | |
| $D_{Vg}$ | Gas phase longitudinal dispersion coefficient | m$^2$ s$^{-1}$ | |
| $D_{Vh}$ | Isothermal vapor conductivity | kg m$^{-2}$ s$^{-1}$ | |
| $D_{VT}$ | Thermal vapor diffusion coefficient | kg m$^{-1}$ s$^{-1}$ °C$^{-1}$ | |
| $h$ | Soil matric potential | m | |
| $H_c$ | Henry's constant | - | 0.02 |
| $K$ | Hydraulic conductivity | m s$^{-1}$ | |
| $K_g$ | Intrinsic air permeability | m$^2$ | |
| $K_{Lh}$ | Isothermal hydraulic conductivities | m s$^{-1}$ | |
| $K_{LT}$ | Thermal hydraulic conductivities | m$^2$ s$^{-1}$ °C$^{-1}$ | |
| $K_s$ | Soil saturated hydraulic conductivity | m s$^{-1}$ | |
| $L_0$ | Latent heat of vaporization of water at the reference temperature | J kg$^{-1}$ | |
| $LAI_{eff}$ | Effective leaf area index | - | |
| $L_f$ | Latent heat of fusion | J kg$^{-1}$ | 3.34E5 |
| $n$ | Van Genuchten fitting parameters | - | |
| $P_g$ | Mixed pore-air pressure | Pa | |
| $q$ | Water flux | kg m$^{-2}$ s$^{-1}$ | |
| $q_a$ | Dry air flux | kg m$^{-2}$ s$^{-1}$ | |
| $q_L$ | Soil liquid water fluxes (positive upwards) | kg m$^{-2}$ s$^{-1}$ | |
| $q_{La}$ | Liquid water flux driven by the gradient of air pressure | kg m$^{-2}$ s$^{-1}$ | |
| $q_{Lh}$ | Liquid water flux driven by the gradient of matric potential | kg m$^{-2}$ s$^{-1}$ | |
| $q_{LT}$ | Liquid water flux driven by the gradient of temperature | kg m$^{-2}$ s$^{-1}$ | |
| $q_V$ | Soil water vapor fluxes (positive upwards) | kg m$^{-2}$ s$^{-1}$ | |
| $q_{Va}$ | Water vapor flux driven by the gradient of air pressure | kg m$^{-2}$ s$^{-1}$ | |
| $q_{Vh}$ | Water vapor flux driven by the gradient of matric potential | kg m$^{-2}$ s$^{-1}$ | |
| $q_{VT}$ | Water vapor flux driven by the gradient of temperature | kg m$^{-2}$ s$^{-1}$ | |
| $r_a^c$ | Aerodynamic resistance for canopy surface | s m$^{-1}$ | |

| Symbol | Description | Units | Value/Definition |
|---|---|---|---|
| $r_a^s$ | Aerodynamic resistance for bare soil | s m$^{-1}$ | |
| $r_{c,min}$ | Minimum canopy surface resistance | s m$^{-1}$ | |
| $r_{l,min}$ | Minimum leaf stomatal resistance | s m$^{-1}$ | |
| $r_s$ | Soil surface resistance | s m$^{-1}$ | |
| $r_{sl}$ | Resistance to molecular diffusion of the water surface | s m$^{-1}$ | 10 |
| $R_n$ | Net radiation | MJ m$^{-2}$ day$^{-1}$ | |
| $R_n^c$ | Net radiation at the canopy surface | MJ m$^{-2}$ day$^{-1}$ | |
| $R_n^s$ | Net radiation at the soil surface | MJ m$^{-2}$ day$^{-1}$ | |
| $S$ | Sink term for transpiration | s$^{-1}$ | |
| $S_a$ | Degree of saturation of the soil air | - | $=1-S_L$ |
| $S_L$ | Degree of saturation in the soil | - | $=\theta_L/\varepsilon$ |
| $S_h$ | Latent heat flux density | W m$^{-3}$ | $=-\rho_w L \partial q_v/\partial z$ |
| $S_p$ | Potential water uptake rate | s$^{-1}$ | |
| $t$ | Time | s | |
| $T$ | Soil temperature | °C | |
| $T_p$ | Potential transpiration | m s$^{-1}$ | |
| $T_r$ | Arbitrary reference temperature | °C | 20 |
| $W$ | Differential heat of wetting | J kg$^{-1}$ | |
| $z$ | Vertical space coordinate (positive upwards) | m | |
| $\alpha$ | Air entry value of soil | m$^{-1}$ | |
| $\alpha(h)$ | Reduction coefficient related to soil water potential | - | |
| $\varepsilon$ | Porosity | - | |
| $\psi$ | Water potential | m | |
| $\lambda_{eff}$ | Effective thermal conductivity of the soil | W m$^{-1}$ °C$^{-1}$ | |
| $\theta$ | Volumetric water content | m$^3$ m$^{-3}$ | |
| $\theta_i$ | Soil ice volumetric water content | m$^3$ m$^{-3}$ | |
| $\theta_L$ | Soil liquid volumetric water content | m$^3$ m$^{-3}$ | |
| $\theta_V$ | Soil vapor volumetric water content | m$^3$ m$^{-3}$ | |
| $\theta_s$ | Volumetric fraction of solids in the soil | m$^3$ m$^{-3}$ | |
| $\theta_a$ | Volumetric fraction of dry air in the soil | m$^3$ m$^{-3}$ | $=\theta_V$ |
| $\theta_{sat}$ | Saturated soil water content | m$^3$ m$^{-3}$ | |
| $\theta_r$ | Residual soil water content | m$^3$ m$^{-3}$ | |
| $\theta_1$ | Topsoil water content | m$^3$ m$^{-3}$ | |
| $\theta_{min}$ | Minimum water content above which soil is able to deliver vapor at a potential rate | m$^3$ m$^{-3}$ | |
| $\rho_a$ | Air density | kg m$^{-3}$ | |
| $\rho_{da}$ | Density of dry air | kg m$^{-3}$ | |
| $\rho_i$ | Density of ice | kg m$^{-3}$ | 920 |
| $\rho_L$ | Density of soil liquid water | kg m$^{-3}$ | 1000 |
| $\rho_s$ | Density of solids | kg m$^{-3}$ | |
| $\rho_V$ | Density of water vapor | kg m$^{-3}$ | |
| $\gamma_W$ | Specific weight of water | kg m$^{-2}$ s$^{-2}$ | |
| $\mu_a$ | Air viscosity | kg m$^{-2}$ s$^{-1}$ | |
| $\tau$ | Light extinction coefficient | - | |

**Notation**

| Parameter | Symbol | Unit | Value |
|---|---|---|---|
| Volumetric water content | $\theta$ | $m^3 \cdot m^{-3}$ | |
| Water flux | $q$ | $kg \cdot m^{-2} \cdot s^{-1}$ | |
| Vertical space coordinate (positive upwards) | $z$ | $m$ | |
| Sink term for transpiration, evaporation | $S$ | $s^{-1}$ | |
| Density of soil liquid water | $\rho_L$ | $kg \cdot m^{-3}$ | 1000 |
| Hydraulic conductivity | $K$ | $m \cdot s^{-1}$ | |
| Water potential | $\psi$ | $m$ | |
| Time | $t$ | $s$ | |
| Heat capacity of the bulk soil | $C_{soil}$ | $J \cdot kg^{-1} \cdot {}^{\circ}C^{-1}$ | |
| Soil temperature | $T$ | ${}^{\circ}C$ | |
| Effective thermal conductivity of the soil | $\lambda_{eff}$ | $W \cdot m^{-1} \cdot {}^{\circ}C^{-1}$ | |
| Latent heat of fusion | $L_f$ | $J \cdot kg^{-1}$ | 3.34E5 |
| Soil ice volumetric water content | $\theta_i$ | $m^3 \cdot m^{-3}$ | |
| Density of water vapor | $\rho_V$ | $kg \cdot m^{-3}$ | |
| Density of ice | $\rho_i$ | $kg \cdot m^{-3}$ | 920 |
| Soil liquid volumetric water content | $\theta_L$ | $m^3 \cdot m^{-3}$ | |
| Soil vapor volumetric water content | $\theta_V$ | $m^3 \cdot m^{-3}$ | |
| Soil liquid water fluxes (positive upwards) | $q_L$ | $kg \cdot m^{-2} \cdot s^{-1}$ | |
| Soil water vapor fluxes (positive upwards) | $q_V$ | $kg \cdot m^{-2} \cdot s^{-1}$ | |
| Isothermal hydraulic conductivities | $K_{Lh}$ | $m \cdot s^{-1}$ | |
| Thermal hydraulic conductivities | $K_{LT}$ | $m^2 \cdot s^{-1} \cdot {}^{\circ}C^{-1}$ | |
| Isothermal vapor conductivity | $D_{Vh}$ | $kg \cdot m^{-2} \cdot s^{-1}$ | |
| Thermal vapor diffusion coefficient | $D_{VT}$ | $kg \cdot m^{-1} \cdot s^{-1} \cdot {}^{\circ}C^{-1}$ | |
| Specific heat capacity of soil solids | $C_s$ | $J \cdot kg^{-1} \cdot {}^{\circ}C^{-1}$ | |
| Specific heat capacity of liquid | $C_L$ | $J \cdot kg^{-1} \cdot {}^{\circ}C^{-1}$ | 4.186 |
| Specific heat capacity of water vapor | $C_V$ | $J \cdot kg^{-1} \cdot {}^{\circ}C^{-1}$ | 1.87 |
| Specific heat capacity of ice | $C_i$ | $J \cdot kg^{-1} \cdot {}^{\circ}C^{-1}$ | 2.0455 |
| Density of solids | $\rho_s$ | $kg \cdot m^{-3}$ | |
| Volumetric fraction of solids in the soil | $\theta_s$ | $m^3 \cdot m^{-3}$ | |

| Description | Symbol | Units | Value |
|---|---|---|---|
| Arbitrary reference temperature | $T_r$ | °C | 20 |
| Latent heat of vaporization of water at the reference temperature | $L_0$ | J kg$^{-1}$ | |
| Differential heat of wetting | $W$ | J kg$^{-1}$ | |
| Liquid water flux driven by the gradient of matric potential | $q_{Lh}$ | kg m$^{-2}$ s$^{-1}$ | |
| Liquid water flux driven by the gradient of matric potential | $q_{LT}$ | kg m$^{-2}$ s$^{-1}$ | |
| Liquid water flux driven by the gradient of air pressure | $q_{La}$ | kg m$^{-2}$ s$^{-1}$ | |
| Water vapor flux driven by the gradient of matric potential | $q_{Vh}$ | kg m$^{-2}$ s$^{-1}$ | |
| Water vapor flux driven by the gradient of matric potential | $q_{VT}$ | kg m$^{-2}$ s$^{-1}$ | |
| Water vapor flux driven by the gradient of air pressure | $q_{Va}$ | kg m$^{-2}$ s$^{-1}$ | |
| Mixed pore-air pressure | $P_g$ | Pa | |
| Specific weight of water | $\gamma_w$ | kg m$^{-2}$ s$^{-2}$ | |
| Transport coefficient for adsorbed liquid flow due to temperature gradient | $D_{TD}$ | kg m$^{-1}$ s$^{-1}$ °C$^{-1}$ | |
| Isothermal vapor conductivity | $D_{Vh}$ | kg m$^{-2}$ s$^{-1}$ | |
| Thermal vapor diffusion coefficient | $D_{VT}$ | kg m$^{-1}$ s$^{-1}$ °C$^{-1}$ | |
| Advective vapor transfer coefficient | $D_{Va}$ | s | |
| Specific heat capacity of dry air | $C_a$ | J kg$^{-1}$ °C$^{-1}$ | 1.005 |
| Liquid water flux | $q_L$ | kg m$^{-2}$ s$^{-1}$ | |
| Vapor water flux | $q_V$ | kg m$^{-2}$ s$^{-1}$ | |
| Dry air flux | $q_a$ | kg m$^{-2}$ s$^{-1}$ | |
| Porosity | $\epsilon$ | - | |
| Density of dry air | $\rho_{da}$ | kg m$^{-3}$ | |
| Degree of saturation in the soil | $S_L$ | - | $=\theta_L/\epsilon$ |
| Degree of air saturation in the soil | $S_a$ | - | $=1-S_L$ |
| Henry's constant | $H_c$ | - | 0.02 |
| Molecular diffusivity of water vapor in soil | $D_e$ | m$^2$ s$^{-1}$ | |
| Intrinsic air permeability | $K_g$ | m$^2$ | |
| Air viscosity | $\mu_a$ | kg m$^{-2}$ s$^{-1}$ | |
| Volumetric fraction of dry air in the soil | $\theta_a$ | m$^3$ m$^{-3}$ | $=\theta_V$ |
| Gas phase longitudinal dispersion coefficient | $D_{Vg}$ | m$^2$ s$^{-1}$ | |
| Soil saturated hydraulic conductivity | $K_s$ | m s$^{-1}$ | |
| Saturated soil water content | $\theta_s$ | m$^3$ m$^{-3}$ | |

| | | | |
|---|---|---|---|
| Residual soil water content | $\theta_r$ | $m^3 \cdot m^{-3}$ | |
| Air entry value of soil | $\alpha$ | $m^{-1}$ | |
| Van Genuchten fitting parameters | $n$ | - | |
| Apparent heat capacity | $C_{app}$ | $J kg^{-1} {}^\circ C^{-1}$ | $= C_{soil} + \rho_i \dfrac{L_f^2}{gT}\dfrac{d}{d\psi}$ |
| Latent heat flux density | $S_h$ | $W m^{-3}$ | $= -\rho_w L \partial q_v / \partial z$ |

**Reference**

Bao, H., Koike, T., Yang, K., Wang, L., Shrestha, M., and Lawford, P.: Development of an enthalpy-based frozen soil model and its validation in a cold region in China, Journal of Geophysical Research: Atmospheres, 121, 5259-5280, 10.1002/2015jd024451, 2016.

Biskaborn, B. K., Smith, S. L., Noetzli, J., Matthes, H., Vieira, G., Streletskiy, D. A., Schoeneich, P., Romanovsky, V. E., Lewkowicz, A. G., Abramov, A., Allard, M., Boike, J., Cable, W. L., Christiansen, H. H., Delaloye, R., Diekmann, B., Drozdov, D., Etzelmüller, B., Grosse, G., Guglielmin, M., Ingeman-Nielsen, T., Isaksen, K., Ishikawa, M., Johansson, M., Johannsson, H., Joo, A., Kaverin, D., Kholodov, A., Konstantinov, P., Kröger, T., Lambiel, C., Lanckman, J.-P., Luo, D., Malkova, G., Meiklejohn, I., Moskalenko, N., Oliva, M., Phillips, M., Ramos, M., Sannel, A. B. K., Sergeev, D., Seybold, C., Skryabin, P., Vasiliev, A., Wu, Q., Yoshikawa, K., Zheleznyak, M., and Lantuit, H.: Permafrost is warming at a global scale, Nature Communications, 10, 264, 10.1038/s41467-018-08240-4, 2019.

Boone, A., Masson, V., Meyers, T., and Noilhan, J.: The Influence of the Inclusion of Soil Freezing on Simulations by a Soil–Vegetation–Atmosphere Transfer Scheme, J Appl Meteorol, 39, 1544-1569, 10.1175/1520-0450(2000)039<1544:tiotio>2.0.co;2, 2000.

Burke, E. J., Jones, C. D., and Koven, C. D.: Estimating the Permafrost-Carbon Climate Response in the CMIP5 Climate Models Using a Simplified Approach, J Clim, 26, 4897-4909, 10.1175/jcli-d-12-00550.1, 2013.

Cheng, G., and Wu, T.: Responses of permafrost to climate change and their environmental significance, Qinghai-Tibet Plateau, Journal of Geophysical Research: Earth Surface, 112, 10.1029/2006JF000631, 2007.

Cuntz, M., and Haverd, V.: Physically Accurate Soil Freeze-Thaw Processes in a Global Land Surface Scheme, Journal of Advances in Modeling Earth Systems, 10, 54-77, 10.1002/2017MS001100, 2018.

De Vries, D. A.: Simultaneous transfer of heat and moisture in porous media, Eos, Transactions American Geophysical Union, 39, 909-916, 10.1029/TR039i005p00909, 1958.

De Vries, D. A.: The theory of heat and moisture transfer in porous media revisited, International Journal of Heat and Mass Transfer, 30, 1343-1350, https://doi.org/10.1016/0017-9310(87)90166-9, 1987.

Dente, L., Vekerdy, Z., Wen, J., and Su, Z.: Maqu network for validation of satellite-derived soil moisture products, Int J Appl Earth Obs Geoinf, 17, 55-65, 10.1016/j.jag.2011.11.004, 2012.

Feddes, R. A., Kowalik, P. J., and Zaradny, H.: Simulation of field water use and crop yield, Centre for Agricultural Publishing and Documentation, Wageningen, the Netherlands, 189 pp., 1978.

Flerchinger, G. N., and Saxton, K. E.: Simultaneous heat and water model of a freezing snow-residue-soil system. I. Theory and development, Transactions of the American Society of Agricultural Engineers, 32, 565-571, 1989.

Gale, M. R., and Grigal, D. F.: Vertical root distributions of northern tree species in relation to successional status, Canadian Journal of Forest Research, 17, 829-834, 10.1139/x87-131, 1987.

Grenier, C., Anbergen, H., Bense, V., Chanzy, Q., Coon, E., Collier, N., Costard, F., Ferry, M., Frampton, A., Frederick, J., Gonçalvès, J., Holmén, J., Jost, A., Kokh, S., Kurylyk, B., McKenzie, J., Molson, J., Mouche, E., Orgogozo, L., Pannetier, R., Rivière, A., Roux, N., Rühaak, W., Scheidegger, J., Selroos, J. O., Therrien, R., Vidstrand, P., and Voss, C.: Groundwater flow and heat transport for systems undergoing freeze-thaw: Intercomparison of numerical simulators for 2D test cases, Adv Water Resour, 114, 196-218, 10.1016/j.advwatres.2018.02.001, 2018.

Guymon, G. L., and Luthin, J. N.: A coupled heat and moisture transport model for Arctic soils, Water Resour Res, 10, 995-1001, 10.1029/WR010i005p00995, 1974.

Hansson, K., Šimůnek, J., Mizoguchi, M., Lundin, L. C., and van Genuchten, M. T.: Water flow and heat transport in frozen soil: Numerical solution and freeze-thaw applications, Vadose Zone J, 3, 693-704, 2004.

Harlan, R. L.: Analysis of coupled heat-fluid transport in partially frozen soil, Water Resour Res, 9, 1314-1323, 10.1029/WR009i005p01314, 1973.

Hinzman, L. D., Deal, C. J., McGuire, A. D., Mernild, S. H., Polyakov, I. V., and Walsh, J. E.: Trajectory of the Arctic as an integrated system, Ecological Applications, 23, 1837-1868, 10.1890/11-1498.1, 2013.

Iijima, Y., Ohta, T., Kotani, A., Fedorov, A. N., Kodama, Y., and Maximov, T. C.: Sap flow changes in relation to permafrost degradation under increasing precipitation in an eastern Siberian larch forest,

Ecohydrology, 7, 177-187, 10.1002/eco.1366, 2014.

485 Jackson, R. B., Canadell, J., Ehleringer, J. R., Mooney, H. A., Sala, O. E., and Schulze, E. D.: A Global Analysis of Root Distributions for Terrestrial Biomes, Oecologia, 108, 389-411, 1996.

Karra, S., Painter, S. L., and Lichtner, P. C.: Three-phase numerical model for subsurface hydrology in permafrost-affected regions (PFLOTRAN-ICE v1.0), Cryosphere, 8, 1935-1950, 10.5194/tc-8-1935-2014, 2014.

490 Kevin, S., Hugues, L., Vladimir, E. R., Edward, A. G. S., and Ronald, W.: The impact of the permafrost carbon feedback on global climate, Environmental Research Letters, 9, 085003, 2014.

Koren, V., Schaake, J., Mitchell, K., Duan, Q. Y., Chen, F., and Baker, J. M.: A parameterization of snowpack and frozen ground intended for NCEP weather and climate models, Journal of Geophysical Research Atmospheres, 104, 19569-19585, 1999.

495 Kroes, J., Van Dam, J., Groenendijk, P., Hendriks, R., and Jacobs, C.: SWAP version 3.2. Theory description and user manual, Alterra, 2009.

Kurylyk, B. L., and Watanabe, K.: The mathematical representation of freezing and thawing processes in variably-saturated, non-deformable soils, Adv Water Resour, 60, 160-177, 10.1016/j.advwatres.2013.07.016, 2013.

500 Li, Q., Sun, S., and Xue, Y.: Analyses and development of a hierarchy of frozen soil models for cold region study, Journal of Geophysical Research Atmospheres, 115, 10.1029/2009JD012530, 2010.

Luo, L., Robock, A., Vinnikov, K. Y., Schlosser, C. A., Slater, A. G., Boone, A., Braden, H., Cox, P., de Rosnay, P., Dickinson, R. E., Dai, Y., Duan, Q., Etchevers, P., Henderson-Sellers, A., Gedney, N., Gusev, Y. M., Habets, F., Kim, J., Kowalczyk, E., Mitchell, K., Nasonova, O. N., Noilhan, J., Pitman, A. J., Schaake,

505 J., Shmakin, A. B., Smirnova, T. G., Wetzel, P., Xue, Y., Yang, Z. L., and Zeng, Q. C.: Effects of frozen soil on soil temperature, spring infiltration, and runoff: Results from the PILPS 2(d) experiment at Valdai, Russia, J Hydrometeorol, 4, 334-351, 10.1175/1525-7541(2003)4<334:EOFSOS>2.0.CO;2, 2003.

Milly, P. C. D.: Moisture and heat transport in hysteretic, inhomogeneous porous media: A matric head-based formulation and a numerical model, Water Resour Res, 18, 489-498, 10.1029/WR018i003p00489, 1982.

510 Niu, G. Y., Yang, Z. L., Mitchell, K. E., Chen, F., Ek, M. B., Barlage, M., Kumar, A., Manning, K., Niyogi, D., and Rosero, E.: The community Noah land surface model with multiparameterization options (Noah-MP): 1. Model description and evaluation with local-scale measurements, Journal of Geophysical Research: Atmospheres, 116, 2011.

Novak, M. D.: Dynamics of the near-surface evaporation zone and corresponding effects on the surface

515 energy balance of a drying bare soil, Agr Forest Meteorol, 150, 1358-1365, https://doi.org/10.1016/j.agrformet.2010.06.005, 2010.

Orgogozo, L., Prokushkin, A. S., Pokrovsky, O. S., Grenier, C., Quintard, M., Viers, J., and Audry, S.: Water and energy transfer modeling in a permafrost-dominated, forested catchment of Central Siberia: The key role of rooting depth, Permafr Periglac Proc, 30, 75-89, 10.1002/ppp.1995, 2019.

520 Painter, S. L.: Three-phase numerical model of water migration in partially frozen geological media: Model formulation, validation, and applications, Comput Geosci, 15, 69-85, 10.1007/s10596-010-9197-z, 2011.

Philip, J. R., and Vries, D. A. D.: Moisture movement in porous materials under temperature gradients, Eos, Transactions American Geophysical Union, 38, 222-232, 10.1029/TR038i002p00222, 1957.

Prunty, L., and Bell, J.: Infiltration Rate vs. Gas Composition and Pressure in Soil Columns Soil Sci Soc Am

525 J, 71, 1473-1475, 10.2136/sssaj2007.0072N, 2007.

Qi, J., Wang, L., Zhou, J., Song, L., Li, X., and Zeng, T.: Coupled Snow and Frozen Ground Physics Improves Cold Region Hydrological Simulations: An Evaluation at the upper Yangtze River Basin (Tibetan Plateau), Journal of Geophysical Research: Atmospheres, 124, 12985-13004, 10.1029/2019jd031622, 2019.

Richards, L. A.: Capillary Conduction of Liquids Through Porous Mediums, Physics, 1, 318, 1931.

530 Saito, H., Šimůnek, J., and Mohanty, B. P.: Numerical Analysis of Coupled Water, Vapor, and Heat Transport in the Vadose Zone, Vadose Zone J, 5, 784-800, 10.2136/vzj2006.0007, 2006.

Schuur, E. A. G., McGuire, A. D., Schädel, C., Grosse, G., Harden, J. W., Hayes, D. J., Hugelius, G., Koven, C. D., Kuhry, P., Lawrence, D. M., Natali, S. M., Olefeldt, D., Romanovsky, V. E., Schaefer, K., Turetsky, M. R., Treat, C. C., and Vonk, J. E.: Climate change and the permafrost carbon feedback, Nature, 520, 171-

535 179, 10.1038/nature14338, 2015.

Su, Z., Wen, J., Dente, L., van der Velde, R., Wang, L., Ma, Y., Yang, K., and Hu, Z.: The Tibetan Plateau observatory of plateau scale soil moisture and soil temperature (Tibet-Obs) for quantifying uncertainties in

coarse resolution satellite and model products, Hydrol Earth Syst Sci, 15, 2303-2316, 10.5194/hess-15-2303-2011, 2011.

540     Su, Z., de Rosnay, P., Wen, J., Wang, L., and Zeng, Y.: Evaluation of ECMWF's soil moisture analyses using observations on the Tibetan Plateau, Journal of Geophysical Research: Atmospheres, 118, 5304-5318, 10.1002/jgrd.50468, 2013.

Subin, Z. M., Koven, C. D., Riley, W. J., Torn, M. S., Lawrence, D. M., and Swenson, S. C.: Effects of Soil Moisture on the Responses of Soil Temperatures to Climate Change in Cold Regions, J Clim, 26, 3139-3158,
545     10.1175/jcli-d-12-00305.1, 2013.

Swenson, S. C., Lawrence, D. M., and Lee, H.: Improved simulation of the terrestrial hydrological cycle in permafrost regions by the Community Land Model, Journal of Advances in Modeling Earth Systems, 4, 10.1029/2012MS000165, 2012.

Touma, J., and Vauclin, M.: Experimental and numerical analysis of two-phase infiltration in a partially
550     saturated soil, Transport in Porous Media, 1, 27-55, 10.1007/bf01036524, 1986.

van de Griend, A. A., and Owe, M.: Bare soil surface resistance to evaporation by vapor diffusion under semiarid conditions, Water Resour Res, 30, 181-188, 10.1029/93wr02747, 1994.

Viterbo, P., Beljaars, A., Mahfouf, J. F., and Teixeira, J.: The representation of soil moisture freezing and its impact on the stable boundary layer, Q J Roy Meteorol Soc, 125, 2401-2426, 1999.

555     Walvoord, M. A., and Kurylyk, B. L.: Hydrologic Impacts of Thawing Permafrost-A Review, Vadose Zone J, 15, 10.2136/vzj2016.01.0010, 2016.

Wang, C., and Yang, K.: A New Scheme for Considering Soil Water-Heat Transport Coupling Based on Community Land Model: Model Description and Preliminary Validation, Journal of Advances in Modeling Earth Systems, 10, 927-950, 10.1002/2017ms001148, 2018.

560     Wang, L., Zhou, J., Qi, J., Sun, L., Yang, K., Tian, L., Lin, Y., Liu, W., Shrestha, M., Xue, Y., Koike, T., Ma, Y., Li, X., Chen, Y., Chen, D., Piao, S., and Lu, H.: Development of a land surface model with coupled snow and frozen soil physics, Water Resour Res, 53, 5085-5103, 10.1002/2017WR020451, 2017.

Wicky, J., and Hauck, C.: Numerical modelling of convective heat transport by air flow in permafrost talus slopes, The Cryosphere, 11, 1311-1325, 10.5194/tc-11-1311-2017, 2017.

565     Yang, K., Chen, Y. Y., and Qin, J.: Some practical notes on the land surface modeling in the Tibetan Plateau, Hydrol Earth Syst Sci, 13, 687-701, 10.5194/hess-13-687-2009, 2009.

Yu, L., Zeng, Y., Su, Z., Cai, H., and Zheng, Z.: The effect of different evapotranspiration methods on portraying soil water dynamics and ET partitioning in a semi-arid environment in Northwest China, Hydrol Earth Syst Sci, 20, 975-990, 10.5194/hess-20-975-2016, 2016.

570     Yu, L., Zeng, Y., Wen, J., and Su, Z.: Liquid-Vapor-Air Flow in the Frozen Soil, Journal of Geophysical Research: Atmospheres, 123, 7393-7415, 10.1029/2018jd028502, 2018.

Yu, L., Zeng, Y., Fatichi, S., and Su, Z.: How vadose zone mass and energy transfer physics affects the ecohydrological dynamics of a Tibetan meadow?, The Cryosphere Discuss, 2020, 1-31, 10.5194/tc-2020-88, 2020.

575     Zeng, Y., Su, Z., Wan, L., and Wen, J.: Numerical analysis of air-water-heat flow in unsaturated soil: Is it necessary to consider airflow in land surface models?, Journal of Geophysical Research: Atmospheres, 116, D20107, 10.1029/2011JD015835, 2011a.

Zeng, Y., Su, Z., Wan, L., and Wen, J.: A simulation analysis of the advective effect on evaporation using a two-phase heat and mass flow model, Water Resour Res, 47, W10529, 10.1029/2011WR010701, 2011b.

580     Zeng, Y., Su, Z., van der Velde, R., Wang, L., Xu, K., Wang, X., and Wen, J.: Blending Satellite Observed, Model Simulated, and in Situ Measured Soil Moisture over Tibetan Plateau, Remote Sensing, 8, 268, 2016.

Zeng, Y. J., and Su, Z. B.: STEMMUS : Simultaneous Transfer of Engery, Mass and Momentum in Unsaturated Soil, ISBN: 978-90-6164-351-7, University of Twente, Faculty of Geo-Information and Earth Observation (ITC), Enschede, 2013.

585     Zhang, M., Wen, Z., Xue, K., Chen, L., and Li, D.: A coupled model for liquid water, water vapor and heat transport of saturated–unsaturated soil in cold regions: model formulation and verification, Environmental Earth Sciences, 75, 10.1007/s12665-016-5499-3, 2016.

Zhao, H., Zeng, Y., Lv, S., and Su, Z.: Analysis of Soil Hydraulic and Thermal Properties for Land Surface Modelling over the Tibetan Plateau, Earth Syst Sci Data Discuss, 2018, 1-40, 10.5194/essd-2017-122, 2018.

590     Zhao, L., Hu, G., Zou, D., Wu, X., Ma, L., Sun, Z., Yuan, L., Zhou, H., and Liu, S.: Permafrost Changes and Its Effects on Hydrological Processes on Qinghai-Tibet Plateau, Bulletin of Chinese Academy of Sciences,

34, 1233-1246, 2019.

Zheng, D., Van der Velde, R., Su, Z., Wang, X., Wen, J., Booij, M. J., Hoekstra, A. Y., and Chen, Y.: Augmentations to the Noah Model Physics for Application to the Yellow River Source Area. Part I: Soil
595 Water Flow, J Hydrometeorol, 16, 2659-2676, 10.1175/JHM-D-14-0198.1, 2015a.

Zheng, D., Van der Velde, R., Su, Z., Wen, J., Booij, M. J., Hoekstra, A. Y., and Wang, X.: Under-canopy turbulence and root water uptake of a Tibetan meadow ecosystem modeled by Noah-MP, Water Resour Res, 51, 5735-5755, 10.1002/2015wr017115, 2015b.

**Tables and Figures**

600 **Table 1. Governing equations for different complexity of water and heat coupling physics (See appendix for notations)**

| Models | Governing equations (water, heat and air) | Number |
|---|---|---|
|  BCM | $\dfrac{\partial \theta}{\partial t} = -\dfrac{\partial q}{\partial z} - S = \rho_L \dfrac{\partial}{\partial z}\left[K\left(\dfrac{\partial \psi}{\partial z} + 1\right)\right] - S$ | (1) |
| | $\underbrace{C_{soil}\dfrac{\partial T}{\partial t} - \rho_i L_f \dfrac{\partial \theta_i}{\partial t}}_{HC} = \underbrace{\dfrac{\partial}{\partial z}\left(\lambda_{eff}\dfrac{\partial T}{\partial z}\right)}_{CHF}$  | (2) |
|  M | $\dfrac{\partial}{\partial t}(\rho_L \theta_L + \rho_V \theta_V + \rho_i \theta_i) = -\dfrac{\partial}{\partial z}(q_L + q_V) - S$  $= -\dfrac{\partial}{\partial z}(q_{Lh} + q_{LT} + q_{vh} + q_{VT}) - S$  $= \rho_L \dfrac{\partial}{\partial z}\left[K_{Lh}\left(\dfrac{\partial \psi}{\partial z} + 1\right) + K_{LT}\dfrac{\partial T}{\partial z}\right] + \dfrac{\partial}{\partial z}\left[D_{Vh}\dfrac{\partial \psi}{\partial z} + D_{VT}\dfrac{\partial T}{\partial z}\right] - S$ | (3) |
| | $\underbrace{\dfrac{\partial}{\partial t}\left[(\rho_s\theta_s C_s + \rho_L\theta_L C_L + \rho_V\theta_V C_V + \rho_i\theta_i C_i)(T - T_r) + \rho_V\theta_V L_0 - \rho_i\theta_i L_f\right] - \rho_L W\dfrac{\partial\theta_L}{\partial t}}_{HC}$  $= \dfrac{\partial}{\partial z}\underbrace{\left(\lambda_{eff}\dfrac{\partial T}{\partial z}\right)}_{CHF} - \dfrac{\partial}{\partial z}[\underbrace{q_V L_0}_{LHF} + \underbrace{q_V C_V(T - T_r)}_{HFV}] - \dfrac{\partial}{\partial z}\underbrace{[q_L C_L(T - T_r)] - C_L S(T - T_r)}_{HFL}$   | (4) |
|  M-AIR | $\dfrac{\partial}{\partial t}(\rho_L\theta_L + \rho_V\theta_V + \rho_i\theta_{ice}) = -\dfrac{\partial}{\partial z}(q_{Lh} + q_{LT} + q_{La} + q_{vh} + q_{VT} + q_{va}) - S$  $= \rho_L\dfrac{\partial}{\partial z}\left[K\left(\dfrac{\partial\psi}{\partial z} + 1\right) + D_{TD}\dfrac{\partial T}{\partial z} + \dfrac{K}{\gamma_w}\dfrac{\partial P_g}{\partial z}\right] + \dfrac{\partial}{\partial z}\left[D_{Vh}\dfrac{\partial\psi}{\partial z} + D_{VT}\dfrac{\partial T}{\partial z} + D_{Va}\dfrac{\partial P_g}{\partial z}\right] - S$ | (5) |
| | $\underbrace{\dfrac{\partial}{\partial t}\left[(\rho_s\theta_s C_s + \rho_L\theta_L C_L + \rho_V\theta_V C_V + \rho_{da}\theta_a C_a + \rho_i\theta_i C_i)(T - T_r) + \rho_V\theta_V L_0 - \rho_i\theta_i L_f\right] - \rho_L W\dfrac{\partial\theta_L}{\partial t}}_{HC}$  $= \dfrac{\partial}{\partial z}\underbrace{\left(\lambda_{eff}\dfrac{\partial T}{\partial z}\right)}_{CHF} - \dfrac{\partial}{\partial z}[\underbrace{q_V L_0}_{LHF} + \underbrace{q_V C_V(T - T_r)}_{HFV} + \underbrace{q_a C_a(T - T_r)}_{HFa}] -$  $\underbrace{\dfrac{\partial}{\partial z}[q_L C_L(T - T_r)] - C_L S(T - T_r)}_{HFL}$    | (6) |
| | $\dfrac{\partial}{\partial t}[\varepsilon\rho_{da}(S_a + H_c S_L)] = \dfrac{\partial}{\partial z}\left[D_e\dfrac{\partial\rho_{da}}{\partial z} + \rho_{da}\dfrac{S_a K_g}{\mu_a}\dfrac{\partial P_g}{\partial z} - H_c\rho_{da}\dfrac{q_L}{\rho_L} + (\theta_a D_{vg})\dfrac{\partial\rho_{da}}{\partial z}\right]$ | (7) |

**Table 2. The adopted average values of soil texture and hydraulic properties at different depths (See appendix for notations)**

| Soil depth (cm) | Clay (%) | Sand (%) | $K_s$ ($10^{-6}$ m s$^{-1}$) | $\theta_{sat}$ (m$^3$ m$^{-3}$) | $\theta_r$ (m$^3$ m$^{-3}$) | $\alpha$ (m$^{-1}$) | $n$ |
|---|---|---|---|---|---|---|---|
| 5-10 | 9.00 | 44.13 | 1.45 | 0.50 | 0.035 | 0.041 | 1.332 |
| 10-40 | 10.12 | 44.27 | 0.94 | 0.45 | 0.039 | 0.041 | 1.362 |
| 40-160 | 5.59 | 65.55 | 0.68 | 0.41 | 0.045 | 0.075 | 1.590 |

605

**Table 3. Comparative statistics values of observed and simulated soil temperature/moisture with three models, with the bold fonts indicating the best statistical performance**

| Experiment | Statistics | Soil temperature (°C) | | | | | Soil moisture (m$^3$ m$^{-3}$) | | | | |
|---|---|---|---|---|---|---|---|---|---|---|---|
| | | 5cm | 10cm | 20cm | 40cm | 80cm | 5cm | 10cm | 20cm | 40cm | 80cm |
| BCM | BIAS | **-0.039** | 0.177 | -0.022 | -1.103 | -0.140 | 0.009 | 0.009 | 0.005 | **0.004** | 0.002 |
| | RMSE | 0.381 | 0.407 | 0.521 | 1.524 | 0.526 | 0.025 | 0.022 | 0.031 | 0.032 | 0.012 |
| ACM | BIAS | -0.183 | 0.093 | **0.001** | -0.956 | **0.027** | **0.000** | 0.004 | 0.001 | 0.005 | **0.001** |
| | RMSE | 0.365 | **0.314** | 0.186 | 1.168 | 0.128 | **0.008** | 0.007 | 0.003 | 0.007 | **0.002** |
| ACM-AIR | BIAS | -0.187 | **0.093** | 0.005 | **-0.953** | 0.029 | -0.001 | **0.004** | **0.001** | 0.005 | 0.001 |
| | RMSE | **0.362** | 0.316 | **0.180** | 1.168 | **0.126** | 0.011 | **0.006** | **0.003** | **0.007** | 0.002 |

610 Note: $BIAS = \frac{\sum_{i=1}^{n}(y_i - \hat{y}_i)}{n}$, $RMSE = \sqrt{\frac{\sum_{i=1}^{n}(y_i - \hat{y}_i)^2}{n}}$, where $y_i$, $\hat{y}_i$ are the measured and model simulated soil temperature/moisture; n is the number of data points.

**(a) Model setup**

[Figure]

**(b) Meteorological Forcing**

[Figure]

Figure 1. (a) Conceptual illustration of the model setup, the surface/bottom boundary conditions, driving forces, and vertical discretization. (b) Half-hourly measurements of meteorological forcing, including air temperature ($T_{atm}$, °C), relative humidity ($HR_{atm}$, %), net radiation (Rn, W m$^{-2}$), wind speed ($U_{wind}$, m s$^{-1}$), and atmospheric pressure ($P_{atm}$, kPa), during the simulation period. Note that dimensions are not draw to scale, models were ran at one-dimensional scale.

615

[Figure]

[Figure]

620

**Figure 2. Comparison of measured (Obs) and estimated time series of soil temperature at various soil layers using Basic Coupled Model (BCM), Advanced Coupled Model (ACM) and Advanced Coupled Model with Air flow (ACM-AIR).**

[Figure]

[Figure]

**Figure 2̶3. Comparison of measured (Obs) and model simulated time series of soil moisture at various soil layers using Basic Coupled Model (BCM), Advanced Coupled Model (ACM) and Advanced Coupled Model with Air flow (ACM-AIR)uncoupled soil physics (unCPLD), coupled water and heat physics (CPLD) and coupled water and heat physics with air flow (CPLD-AIR) model.**

[Figure]

**Figure 3.** Comparison of measured (Obs) and model simulated freezing front propagation (FFP) using  Basic Coupled Model (BCM),  Advanced Coupled Model (ACM) and Advanced Coupled Model with Air flow (ACM-AIR). Note the measured FFP was seen as the development of zero degree isothermal lines from the measured soil temperature field.

635

[Figure]

**Figure 45.** Scatter plot of observed and model estimated half-hourly latent heat flux using (a) uncoupled soil physics (unCPLD), (b) coupled water and heat physics (CPLD) and (c) coupled water and heat physics with air flow (CPLD-AIR) model. The color indicates the data composite of surface latent heat flux.

640

[Figure]

**Figure** 6. Comparison of observed and model simulated (a) mean diurnal variations of surface evapotranspiration and (b) cumulative evapotranspiration (ET) by Basic Coupled Model (BCM), Advanced Coupled Model (ACM), and Advanced Coupled Model with Air flow (ACM-AIR) .

[Figure]

**Figure 67.** Time series of model simulated heat budget components at the soil depth of 5cm using (a &d) Basic Coupled Model (BCM)unCPLD, (b &e) Advanced Coupled Model (ACM)CPLD, and (c &f) Advanced Coupled Model with Air flow (ACM-AIR)CPLD-AIR simulations during the typical 6-day freezing (left column) and freezing-thawing transition (right column) periods. HC, rate of change of heat content, CHF, conductive heat flux divergence, HFL, convective heat flux divergence due to liquid water flow, HFV, convective heat flux divergence due to water vapor flow, HFa, convective heat flux divergence due to air flow, LHF, latent heat flux divergence. Note that for graphical purposes, HFL, HFV, HFa, and LHF were enhanced by a factor of 10 during the freezing period.

[Figure]

**Figure 8.** The spatial and temporal distributions of model estimated soil latent heat flux density using (a &d) Advanced Coupled Model (ACM), (b &e) Advanced Coupled Model with Air flow (ACM-AIR) and (c &f) the difference between ACM and ACM-AIR simulations ($S_{h,\textit{CPLD}\,\underline{ACM}-AIR} - S_{h,\textit{CPLD}\,\underline{ACM}}$) during the typical 6-day freezing and freezing-thawing transition periods. The left and right column are for the freezing and freezing-thawing transition period, respectively. Note that figures for the Basic Coupled Model (BCM) are absent as it can not simulate the subsurface soil latent heat flux density.